# Subglacial water amplifies Antarctic contributions to sea-level rise

Chen Zhao [1,2,8] ✉, Rupert Gladstone [3], Thomas Zwinger [4], Fabien Gillet-Chaulet [5], Yu Wang[1], Justine Caillet [5], Pierre Mathiot [5], Leopekka Saraste [4], Eliot Jager [6], Benjamin K. Galton-Fenzi [1,2,7], Poul Christoffersen [1,2,8] & Matt A. King [2,9]

Antarctica's contribution to global sea-level rise is deeply uncertain, with subglacial water suspected to play a critical role, yet its impact remains unclear. We demonstrate that water at the base of ice sheets influences sliding behaviour and that its exclusion from models can underestimate sea-level rise projections and delay the predicted onset of tipping points. Here we use an Antarctic Ice Sheet model (Elmer/Ice) to explore how different assumptions about water pressure at the ice base affect sea-level rise projections from 2015 to 2300. Our results indicate that incorporating subglacial water can amplify ice discharge across the Antarctic Ice Sheet by up to threefold above the standard approach, potentially contributing an additional 2.2 metres to sea-level rise by 2300. Notably, a smoothly decreasing basal drag near the grounding line more than doubles grounding line flux by 2300 relative to scenarios where effective pressure is simplified into a spatially constant coefficient. Basin-specific responses vary significantly, with some scenarios advancing tipping points by up to 40 years. These findings underscore the critical need to integrate evolving subglacial hydrology into ice sheet models.

The Antarctic Ice Sheet provides the greatest source of uncertainty in current sea-level rise projections[1,2]. This deep uncertainty, and the risk of potential tipping points that can cause rapid ice loss, is driven by the lack of understanding about fundamental processes influencing ice-sheet dynamics[3]. In the Ice Sheet Model Intercomparison for CMIP6 (ISMIP6), more than half of the total uncertainty is due to discrepancies between different ice-sheet simulations[4,5]. While it is challenging to model the complex interactions of Antarctic Ice Sheet with climate[4,6], a key process that remains largely unconstrained by observations is basal sliding.

The widespread production of subglacial water beneath the Antarctic Ice Sheet, and the potential intrusion of warm seawater near the grounding line, plays a crucial role in modulating basal sliding and enabling rapid ice flow[7–11]. Subglacial water drainage beneath the ice sheet occurs through two primary pathways: distributed flow and channelized flow[12]. The existence of channelized drainage beneath the Antarctic Ice Sheet has been inferred from the alignment of predicted subglacial meltwater flux across the grounding line with observed 'subglacially-sourced' sub-ice-shelf channels[13–16]. These drainage channels are fed by a distributed drainage system, which consists of inter-linked cavities and serves as an initially inefficient pathway for subglacial meltwater transport[12]. The source of the distributed and channelised subglacial hydrologic systems is meltwater production

[1]Australian Antarctic Program Partnership, Institute for Marine and Antarctic Studies, University of Tasmania, Hobart, Australia. [2]Australian Centre for Excellence in Antarctic Science, University of Tasmania, Hobart, Australia. [3]Arctic Centre, University of Lapland, Rovaniemi, Finland. [4]CSC-IT Center for Science, Espoo, Finland. [5]Univ. Grenoble Alpes, CNRS, INRAE, IRD, Grenoble INP, Institut des Géosciences de l'Environnement, Grenoble, France. [6]Institute for Atmospheric and Earth System Research / Physics, Faculty of Sciences, University of Helsinki, Helsinki, Finland. [7]Australian Antarctic Division, Kingston, Australia. [8]Institute for Marine and Antarctic Studies, University of Tasmania, Hobart, Australia. [9]School of Geography, Planning, and Spatial Sciences, University of Tasmania, Hobart, Australia. ✉ e-mail: chen.zhao@utas.edu.au

from both frictional heating, due to basal sliding and internal ice deformation, and geothermal heating from the Earth's interior[7].

The Antarctic Ice Sheet flows through two primary mechanisms: internal ice deformation and sliding over the bed, with the latter dominating the dynamics of all fast flowing outlet glaciers. Accurately representing basal sliding is essential for projecting future ice sheet change. The processes controlling sliding broadly fall into two regimes: The 'Weertman' regime, represented in models by expressing basal resistance as a power law function of sliding speed[17], and the 'Coulomb' regime, represented in models by expressing basal resistance as a function of the effective normal force at the ice-bedrock interface[18,19]. For ice sheets, this normal force is given by the ice overburden pressure minus the basal water pressure, henceforth referred to as 'effective pressure', $N$. The Weertman regime is expected to apply for the case of hard bedrock, low sliding speeds, and high effective pressure. The Coulomb regime is expected to apply for the case of a soft (sediment) bed, fast sliding speeds, and low effective pressure. Recent laboratory experiments[20] have yielded compelling evidence that a 'regularised Coulomb' (hereafter 'RC') sliding parameterisation, capturing both Weertman and Coulomb regimes, provides a unified framework for modelling sliding across diverse bed types[18,21–23]. When applying RC sliding, the regime shift between Weertman and Coulomb type sliding is governed by sliding speed, effective pressure, and parameter choices.

Because the actual distribution of effective pressure under the Antarctic Ice Sheet is unknown, model-based sea-level rise projections typically either apply Weertman sliding everywhere, which neglects effective pressure entirely (e.g.,refs. [24–26]) or make simple assumptions about effective pressure when applying RC sliding (e.g., refs. [27–29]). A summary of sliding laws used in three major Model Intercomparison Projects (Table S1) reveals that the majority of Antarctic Ice Sheet models (e.g., 10 of 16 in ISMIP6-2100[2], 10 of 19 in ISMIP6-2300[5], and 7 of 15 in ABUMIP[30]) employed the Weertman sliding relation.

For models employing RC sliding, assumptions must be made about effective pressure. One such assumption is perfect hydrological connectivity between the subglacial system and the ocean[27–29,31]. Another one is to prescribe effective pressure as proportional to ice overburden pressure[32,33]. While practical, these approaches lack robust observational support and may lead to systematic over- or underestimation of basal water pressures across large regions[27,31,34–36]. An alternative approach with a stronger physical justification is to simulate effective pressure using more advanced hydrology models. Models such as GlaDS (Glacier Drainage System[37]) and SHAKTI[38] offer distributed and channelized drainage systems that can capture spatial variability in effective pressure. While these models improve physical realism, they are computationally expensive and sensitive to uncertainties in basal boundary conditions (e.g., bed roughness, geothermal heat flux)[35,39].

A simpler version of RC sliding has been proposed that removes explicit dependence on effective pressure[21,40]. In this approach, effective pressure is instead implicitly embedded within spatially optimized parameters of the sliding relation, which are calibrated to reproduce observed ice flow speeds[21]. While computationally efficient and practical for large-scale ice sheet simulations, this method may overlook important spatial and temporal variability in effective pressure.

A further assumption about the effective pressure and its impact on sliding near the grounding line can be imposed using a 'height-above floatation' (HAF) scaling[21]. HAF is defined as the height of the upper ice surface above sea level minus the height required for the ice overburden pressure to match the ocean water pressure at the bed. This assumption scales the basal resistance or the effective pressure from a prescribed HAF value down to zero at the grounding line, reflecting the influence of warm seawater intrusion beneath grounded ice[41]. However, the inland extent of this effect is poorly constrained due to limited observations, and this approach neglects significant spatial and temporal variability driven by evolving glacier geometry[42,43]. Both the HAF-scaling and perfect ocean connection assumption enforce the floating condition at the grounding line. However, HAF-scaling, applied only near the grounding line, adjusts basal resistance based on ice elevation relative to flotation, potentially under- or overestimating effective pressure. In contrast, the perfect ocean connection assumes uniform hydraulic connectivity across the domain, systematically overestimating effective pressure by neglecting local hydraulic potential gradients.

Building on these approaches, we conduct six experiments (Table 1) to assess the impact of these differing assumptions about spatial and temporal variations in effective pressure on projections of future Antarctic Ice Sheet behaviour. These experiments include approaches with both implicit and explicit representations of effective pressure, including the assumption of perfect ocean connection, hydrology model-derived estimates, and height-above-flotation (HAF) scaling. For comparison against the more physically justifiable RC sliding parameterisation, we also apply the linear Weertman sliding relation (hereafter 'LW'), which, despite its tendency to underestimate mass loss[21], is still widely used for its simplicity in both numerical and computational aspects[2,5,26,30]. By isolating these parameterisations, we evaluate their influence on basal sliding and the resulting ice dynamics, providing insights into the role of hydrology in ice sheet modelling.

## Results
### Subglacial effective pressure must be included in sliding relations
Our simulations show large differences in projected total Antarctic ice-mass loss between experiments. The largest differences are between LW and the RC relation with implicit effective pressure and HAF-scaling (**RC_iN_HAF**; Fig. 1). By 2300, the grounding line flux using **RC_iN_HAF** (8453 Gt yr⁻¹) is almost four times larger than using LW (2245 Gt yr⁻¹) under a high emission scenario. While **LW** suggests a negative sea-level contribution, the **RC** simulations all show a positive sea-level contribution by 2300, with **RC_iN_HAF** producing the largest rise of ~2 m sea-level equivalent (SLE; Fig. 1b).

The LW relation, which ignores the role of subglacial water pressure, underestimates sea-level rise contributions relative to the RC

**Table 1 | Summary of sliding parameterisations and effective pressure treatments**

| Experiment | Effective Pressure, $N$ | Description | HAF Scaling[a] |
|---|---|---|---|
| LW | None | Sliding independent of effective pressure. | No |
| RC_iN | Implicit | $N$ subsumed into a constant sliding coefficient. | No |
| RC_iN_HAF | Implicit | $N$ subsumed into a constant sliding coefficient. | Yes |
| RC_eN_POC | Explicit | Perfect ocean connection assumption for $N$. | No |
| RC_eN_GlaDS | Explicit | $N$ simulated using a distributed hydrology model. | No |
| RC_eN_GlaDS_HAF | Explicit | $N$ simulated using a distributed hydrology model. | Yes |

[a]HAF Scaling Height-above-flotation scaling.

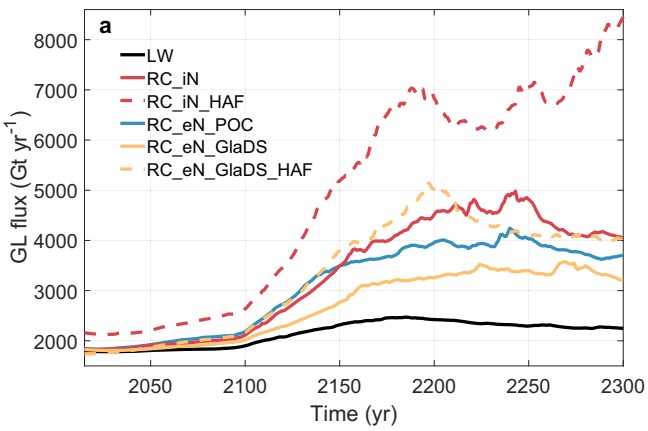
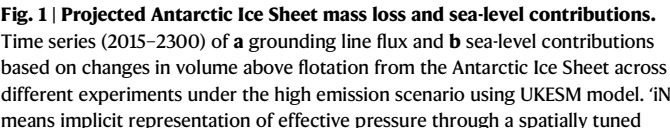
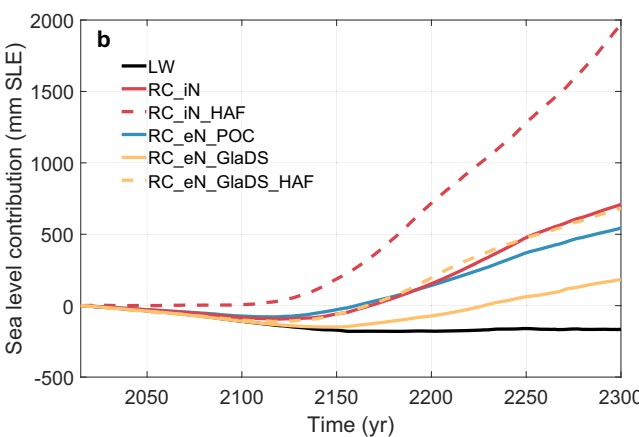

**Fig. 1 | Projected Antarctic Ice Sheet mass loss and sea-level contributions.** Time series (2015–2300) of **a** grounding line flux and **b** sea-level contributions based on changes in volume above flotation from the Antarctic Ice Sheet across different experiments under the high emission scenario using UKESM model. 'iN' means implicit representation of effective pressure through a spatially tuned

parameter while 'eN' means explicit representation of effective pressure. 'HAF' means height-above flotation scaling. GlaDS means simulated effective pressure from the GlaDS hydrology model, while POC means effective pressure is calculated based on perfect ocean connection assumption.

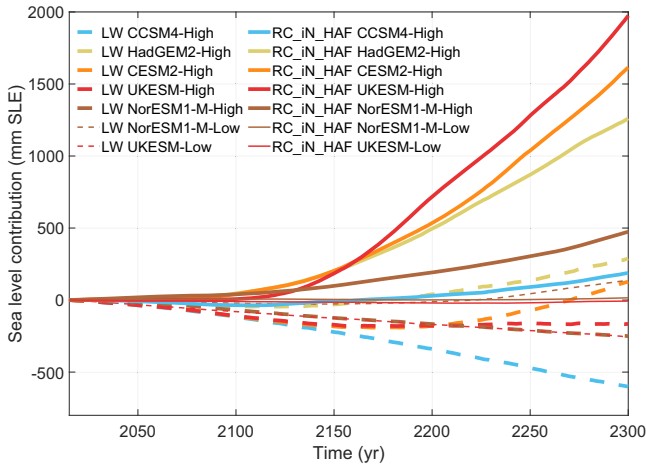

**Fig. 2 | Time series (2015–2300) of sea-level contributions in experiment using linear Weertman law (dashed lines) and regularised Coulomb law (solid lines) under various climate models.** Thick and thin lines represent high and low emission scenarios, respectively. The sea-level contributions are determined by the changes in volume above flotation relative to 2015.

relation under both low and high emission scenarios (Fig. 2). Under high emission scenarios, all RC experiments produce a positive sea-level contribution, whereas this is the case for only two LW experiments. Under the two low emission scenarios, the RC relation suggests near equilibrium by 2300, which aligns more closely with observed mass loss in recent decades[44], as opposed to the substantial mass gain (UKESM) or loss (NorESM1-M) with the LW relation (see further analysis in the Supplementary Material).

Substantial variations of sea-level rise contributions are produced in each Antarctic sector depending on the choice of sliding relation used (Fig. 3). The West Antarctic Ice Sheet exhibits the largest differences. For West Antarctica, under a high emission scenario and using the RC relation, we obtain a net mass loss of 6–24 mm SLE by 2100 for all climate models, except CCSM4, which predicts a mass gain. Conversely, with the LW relation, three of five climate models produce a mass gain. Beyond 2100, both sliding relations produce positive sea-level rise contributions from West Antarctica. However, using the RC relation results in contributions up to five times higher than using the LW relation by 2300. For the East Antarctic Ice Sheet, all climate

models predict net mass gain for this century, with LW predicting larger numbers than RC. However, East Antarctica starts to lose more ice after 2100 using the RC relation, with sea-level rise reaching 17–93 mm by 2200 and 23–248 mm by 2300. In contrast, LW produces overall mass gain, with three of five the climate models producing negative sea-level rise over the period 2200–2300. The Antarctic Peninsula shows less sensitivity to the choice of sliding relations, due to the steep valley glaciers and low water pressure in this region, where the high driving stress means the ice flow is relatively insensitive to the choice of sliding relation.

The RC simulations, incorporating either an explicit or implicit dependence on effective pressure, consistently predict higher mass loss than LW under strong warming scenarios, and exhibit a wide spread in sea-level projections due to differing assumptions about effective pressure distribution. These results highlight the importance of representing effective pressure in Antarctic Ice Sheet projections and carefully evaluating the assumptions about its spatial distribution.

## Subglacial effective pressure controls ice fluxes

Our simulations reveal the treatment of effective pressure significantly affects the spatial tuning of sliding coefficients, which, in turn, impacts the evolution of basal shear stress. For example, at the continental-scale, the implicit representation of effective pressure produces higher grounding line flux and mass loss than its explicit treatment (Fig. 1). A negative sea-level rise contribution before 2100 is simulated in most RC experiments (Fig. 1b), which is primarily due to high surface mass inputs from the climate models (see Supplementary Material), however, the ice dynamic loss characterised by grounding line flux is still positive in this century ( ~2000 Gt yr$^{-1}$).

The grounding line flux in the implicit run (RC_iN) increases after 2100, peaking at 4984 Gt yr$^{-1}$ around 2250. However, when HAF-scaling is introduced (RC_iN_HAF), the results diverge dramatically, showing exponential growth in grounding line flux and reaching over 8000 Gt yr$^{-1}$ – the highest ice discharge across the grounding line– displaying a significant disparity in magnitude. This suggests that HAF-scaling enhances basal sliding due to the progressive reduction of basal shear stress over time as the ice approaches flotation. The corresponding sea-level rise contribution reaches nearly 2 m by 2300, almost three times the increase relative to the non-HAF-scaling case (RC_iN). The divergence between the two curves beyond 2100 highlights how poorly constrained basal shear stress near the grounding line can significantly alter long-term sea-level rise projections.

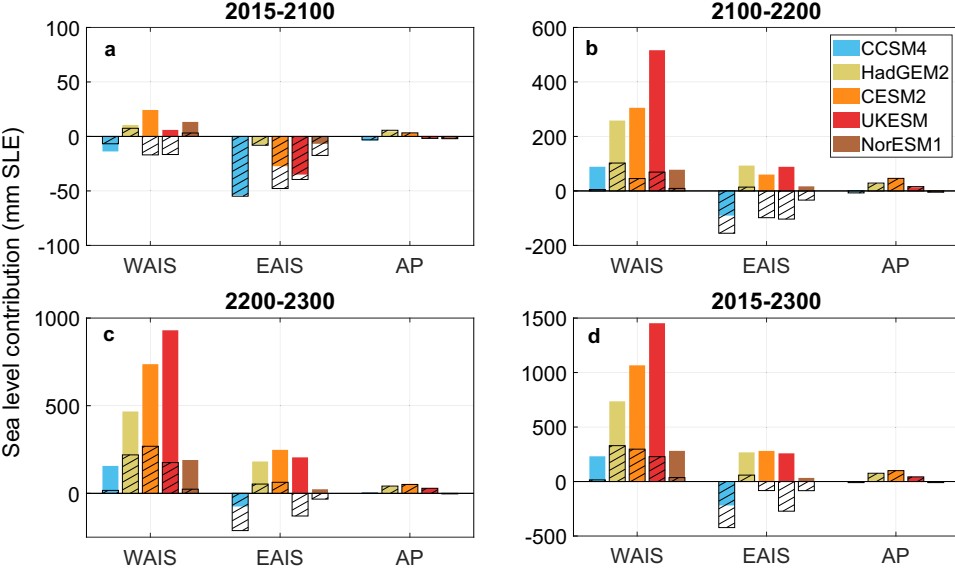

**Fig. 3 | Sea-level contributions from various sectors of Antarctica across different periods.** Sea-level contributions (sea level equivalent, SLE) from West Antarctic (WAIS), East Antarctic (EAIS), and Antarctic Peninsula (AP) from regularised Coulomb with implicit effective pressure and HAF-scaling (RC_iN_HAF; filled) and linear Weertman (hachured) sliding relations over **a** 2015–2100, **b** 2100–2200, **c** 2200–2300, and **d** 2015–2300. Different colours represent various climate models with high emission scenarios.

Under the explicit representation of effective pressure, the perfect ocean connection assumption has been commonly used in previous regional and continental-scale studies[27–29], due to limited observations and the less advanced state of hydrology models. Under such assumption, RC_eN_POC updates effective pressure over time, which is more realistic near the grounding line where the ice approaches flotation, although this approach systematically underestimates basal water pressure which is not only influenced by the sea level but also the subglacial hydrological system (Fig. S11). While this parameterisation ensures the Coulomb regime dominates near the grounding line, the calculated effective pressure based on ice geometries is consistently higher as compared with simulations using the hydrology model (see Supplementary Material), resulting in the second lowest mass loss (540 mm SLE). The grounding line flux increases from 2185 Gt yr⁻¹ to 3698 Gt yr⁻¹, with a peak flux of ~4245 Gt yr⁻¹ at the year 2240.

While we adopted the simulated effective pressure from the GlaDS hydrology model, the sole experiment enforcing a constant effective pressure (RC_eN_GlaDS) produces the least multi-century ice-mass loss (183 mm SLE by 2300 in Fig. 1b), with the grounding line flux increasing from 2012 Gt yr⁻¹ to 3209 Gt yr⁻¹, with a later and lower peak at ~3578 Gt yr⁻¹ around the year 2270. Similar to the comparison between RC_iN and RC_iN_HAF, with a HAF-scaling applied to the constant effective pressure from GlaDS (RC_eN_GlaDS_HAF), the grounding line flux matches RC_eN_POC by the year 2140, and subsequently undergoes a rapid increase, peaking at 5152 Gt yr⁻¹ around the year 2200. The sea-level rise contribution reaches 682 mm SLE, nearly four times of the case with a constant effective pressure.

The assumption made about effective pressure substantially impacts the timing of tipping points (Fig. 1a), whereby peaks in grounding line flux correspond to rapid grounding line retreat along a retrograde slope. Both experiments with HAF-scaling reach a maximum grounding line flux around 2200, at least 40 years earlier than other experiments. The flux in the explicit HAF-scaled experiment with GlaDS gradually decreases after 2200, whereas the implicit treatment of effective pressure in RC_iN_HAF produces an additional maximum around 2250 which continues increasing by 2300. The modification of effective pressure through HAF-scaling substantially amplifies the grounding line flux. This underscores the significance of the magnitude and spatial distribution of effective pressure near the grounding line, and subsequent impact on ice sheet discharge and sea-level rise.

Almost all basins exhibit the highest mass loss from RC_iN_HAF and lowest from LW (Fig. 4). Four basins with reverse bed slopes, that is G-H (feeding Pine Island and Thwaites Glaciers), E-EP (Ross Ice Shelf), D-Dp (Cook Glacier) and Cp-D (Totten Glacier), are chosen for a closer analysis because these are especially vulnerable to marine ice sheet instability (MISI)[45]. In West Antarctica, Basin G-H, feeding Pine Island Glacier (PIG) and Thwaites Glacier, has experienced significant grounding line retreat and is particularly sensitive to the MISI[25,46]. Most experiments suggest a gradual increase in ice flux to 275–370 Gt yr⁻¹ by 2300. With implicit effective pressure and HAF-scaling, the projection is nearly 10 times larger than other parameterisations. In this region, the grounding line at 2300 retreats more than 300 km inland compared to its initial position in 2015 (Fig. 5c), while other experiments show a retreat of about 60 km. Notably, the grounding line retreats over 100 km in just 10 years around 2280 (Fig. 5c). The timing of reaching a tipping point varies, with RC_eN_POC reaching the tipping point around 2095, almost 100 years earlier than RC_eN_GlaDS_HAF (2185) and LW (2215). In contrast, for Basin Ep-F, which feeds the Ross Ice Shelf, the total grounding line flux and movement is relatively invariant across all experiments.

In East Antarctica, the Wilkes Subglacial Basin (Basin D-Dp) projection with implicit effective pressure and HAF-scaling reaches its first tipping point with a peak grounding line flux of 503 Gt/yr by 2190, a factor of 1.6 larger and nearly a century earlier than the experiment with effective pressure calculated based on evolving ice geometry (RC_eN_POC). The Recovery basin (Basin Jpp-K) reaches its first tipping point around 2200 across all experiments, but another tipping point is reached around 2260 in RC_iN_HAF that is two times larger than the other RC experiments. In contrast, the Aurora Subglacial Basin (Basin Cp-D) is less sensitive, with peaks between 2170 and 2190 and a lower rate of mass loss until 2300. Separate research on this basin shows that ice-hydrology interaction increases sea-level contribution by 30% at 2100, compared to an ice-only model with constant effective pressure from a hydrology model[9]. This aligns with the comparison presented here between the two experiments adopting effective pressure from GlaDS, where HAF-scaling enhances the grounding line flux by approximately 20%.

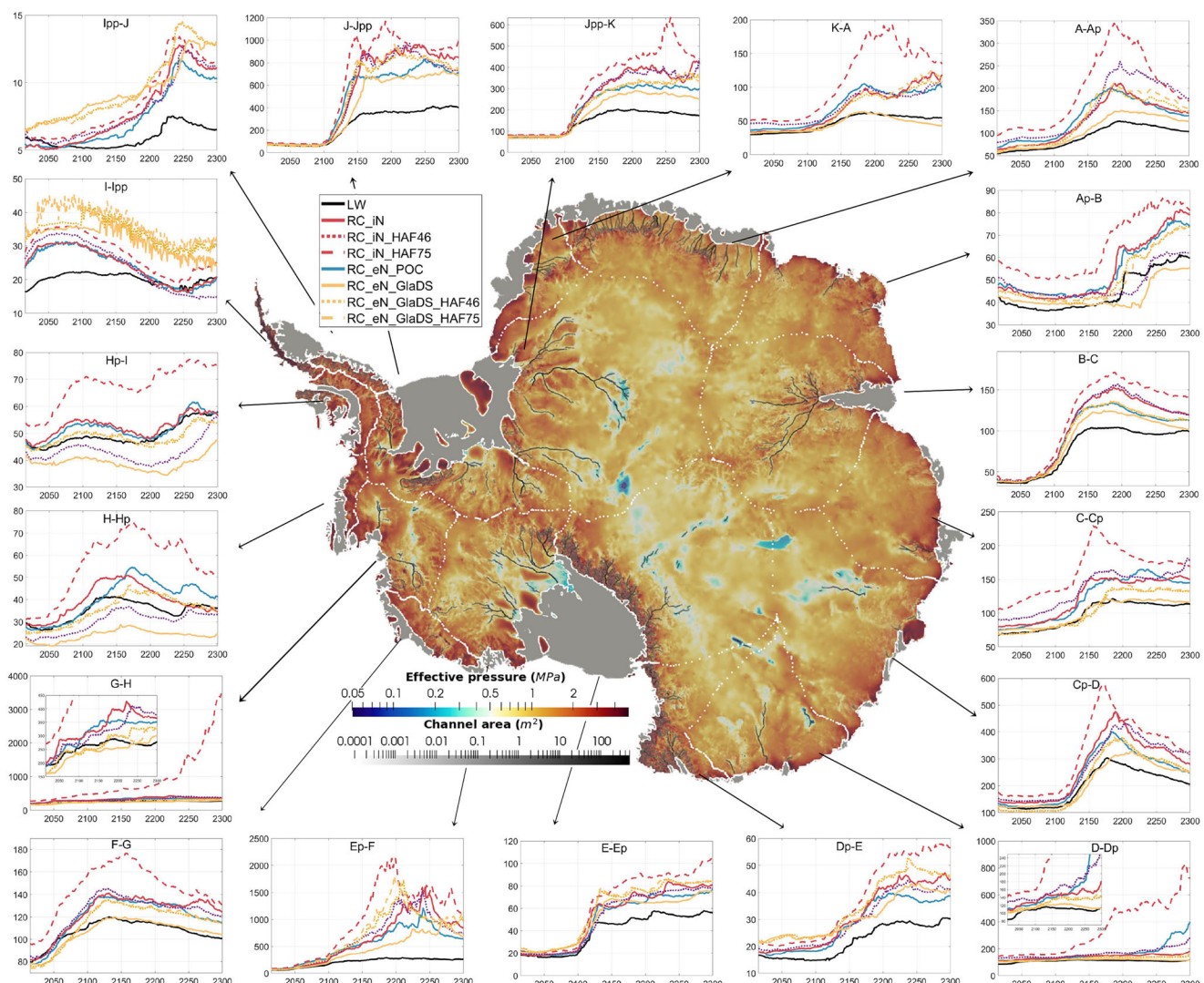

**Fig. 4 | Time series (2015–2300) of grounding line flux (Gt yr⁻¹) across various basins under different experiments.** The background image is the simulated effective pressure and channel area (cross-sectional area of each channel) generated from GlaDS. *Y* axis is the grounding line flux (unit: Gt yr⁻¹) and *X* axis is the time (unit: year). The Antarctic drainage basins are from E. Rignot and J. Mouginot[65].

## Increased sensitivity to effective pressure near the grounding line

To better understand how assumptions about effective pressure can lead to such a wide spread of sea-level rise projections for RC sliding, we examine the physical processes near the grounding line. As subglacial water crosses the grounding line into the ocean, it reaches pressure balance with the ocean, driving the effective pressure at the bed near the grounding line to nearly zero. This results in a Coulomb regime with low basal resistance, with an upstream regime shift toward Weertman sliding. As the grounding line retreats, this low-drag region migrates upstream, allowing further speed up of ice flow, providing a positive feedback enhancing grounding line retreat and ice sheet mass loss. Grounding line retreat also exposes more of the ice shelf to ocean-induced melting, further accelerating the process. In regions with low gradients in bedrock elevation and ice thickness, such as the Siple Coast feeding the Ross Ice Shelf, this low-drag Coulomb regime can extend far from the grounding line (Fig. S12c), forming an extensive ice plain[47].

All RC implementations in this study (see Methods), except for the later stages of the RC_eN_GlaDS simulation (after significant grounding line retreat), impose a low-drag Coulomb region near the grounding line due to assumptions about the spatial distribution of effective pressure, either explicitly (e.g., the perfect ocean connection assumption) or implicitly (e.g., through the HAF-scaling). These differing assumptions lead to variations in the location of the Weertman/Coulomb regime shift and the extent of the low-drag region. For example, the threshold $h_T$ in HAF-scaling determines the extent of basal drag reduction upstream of the grounding line as a function of ice geometry[48]. Separate tests (see Supplementary Material) using a lower $h_T$ value (46 m) underscore the sensitivity of the low-drag region's extent to this threshold, highlighting its importance in sea-level rise projections and raising concerns about the broader validity of this approach.

Taking Thwaites Glacier as an example, the RC_iN_HAF simulation exhibits large-scale collapse during the 23rd century, whereas the other simulations show a more modest retreat. While changes in basal drag before 2015 are relatively small (Fig. S14d), the area of low basal drag in 2270 for RC_iN_HAF extends more than 50 km inland (Fig. S15d), much farther than in the other simulations (typically 10–15 km upstream of the grounding line). This enlarged low-drag region, which represents the formation of a near-flotation ice plain, is conditioned by the $h_T$ parameter as well as the bedrock geometry of the region into which the grounding line has retreated. The existence of such a zone of low basal drag explains the significantly high mass loss for Thwaites Glacier in this simulation. A similar low-drag ice plain exists at the ice streams feeding the Ross Ice Shelf[47], which is also a sensitive region in our model (Fig. S4).

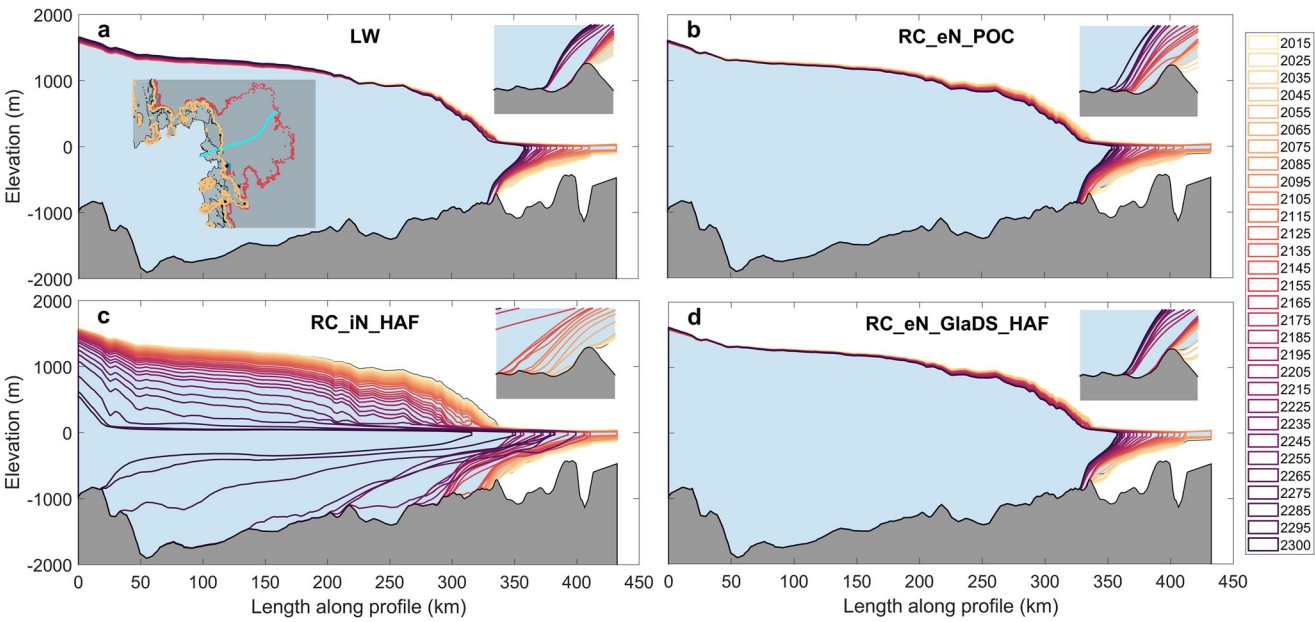

**Fig. 5 | Time series (2015–2300) of ice-geometries along a Thwaites Glacier flowline.** Ice-geometries along the flowline (cyan colour in the inset) of Thwaites Glacier in Basin G-H, indicating the grounding line movement under different experiments: **a** linear Weertman (LW), **b** RC_eN_POC, **c** RC_iN_HAF, and **d** RC_eN_GlaDS_HAF. The coloured dots in the inset of (**a**) show the grounding line position in 2300 from LW (light grey), RC_iN_HAF (purple), RC_eN_POC (blue), and RC_eN_GlaDS_HAF (red).

The primary difference between implicit and explicit representations of effective pressure lies in the location where the shift from the Weertman regime (slower inland flow) to the Coulomb regime (faster outlet flow) occurs, which determines the extent of the low-drag region or ice plain. The Coulomb limit defines the maximum basal shear stress under a Coulomb sliding regime, where sliding is primarily controlled by effective pressure and the sliding coefficient. In simulations using implicit representations of effective pressure (RC_iN and RC_iN_HAF), fast-flowing regions where basal sliding velocity exceeds 300 m yr$^{-1}$ exhibit Coulomb-plastic behaviour (see Methods for more details), effectively marking the Coulomb limit. Conversely, simulations with explicit effective pressure representation determine the Coulomb limit via the term $C^m N^m A_s$, which varies spatially and temporally with effective pressure. As a result, experiments with implicit effective pressure produce much larger Coulomb-dominated regions compared to those with explicit effective pressure, where the Coulomb regime is confined to less than 10 km upstream of the moving grounding line (Fig. S12). This distinction explains why RC_iN_HAF exhibits greater sensitivity to HAF-scaling compared to simulations that incorporate effective pressure from a hydrology model (RC_eN_GlaDS_HAF).

## Discussion

Our findings emphasize the critical role of subglacial water pressure and how a relatively small region close to the grounding line drives multi-century ice sheet mass loss. This aligns with the findings by ref. 11, which demonstrate that decreased effective pressure near the grounding line amplifies the ice sheet's sensitivity to climatic forcing, particularly for a given sliding relation power. Approaching the Coulomb limit and zero friction near the grounding line allows a more mobile grounding line with substantial mass loss potential. When this transition happens over a narrow area, model resolution becomes important, as ice-sheet models often struggle with small-scale transitions from high to zero drag near the grounding line[31]. We also find a strong sensitivity of ice sheet projections to assumptions about the spatial distribution of effective pressure, particularly to threshold parameters that govern the Weertman to Coulomb regime shift.

Previous studies[21,49] show that model projections based on LW cannot reproduce observations as well as RC that incorporate

sub-hydrology changes as part of the parameterisation. Among these RC experiments, we place greater confidence in the results from experiment RC_eN_GlaDS_HAF, as it uses a more physically realistic subglacial hydrology system where HAF-scaling adjusts effective pressure as the grounding line retreats. However, while the HAF-scaling accounts for some of these time-varying effects, the entire hydraulic regime may have shifted, particularly in regions experiencing significant changes in ice flow or grounding line dynamics[12,39]. Moreover, the GlaDS model, originally designed to simulate subglacial hydrology on hard bedrock, may predict more efficient drainage systems than those occurring on a soft bed, potentially leading to lower modelled subglacial water pressures[12,37,39]. This highlights the importance of accurately predicting subglacial water pressure and developing a more comprehensive ice sheet-hydrology model that can account for the temporal co-evolution of ice sheet dynamics and the subglacial hydrological system.

Our simulations explore the sensitivity to basal sliding parameterisations under various climate scenarios. However, limitations in our approach highlight areas for improvement. The use of SSA assumes vertical hydrostatic equilibrium and neglects vertical shear stresses, which restricts its ability to fully capture the ice dynamics at the grounding line. This is particularly relevant in regions with pronounced bedrock slopes, where non-hydrostatic effects and deviatoric (bridging) stresses-ignored by SSA become significant. While a fully resolved full-Stokes model, combined with a contact problem formulation for the lower ice surface, would be needed to consistently resolve these dynamics[50], we argue that the sliding relation itself is less sensitive to the choice of stress representation in the ice-flow model. For instance, a similar sensitivity to driving stress has been found between an SSA model and one incorporating simple vertical shear, particularly under a regularized Coulomb or Weertman sliding parameterisation[21].

Nevertheless, unresolved processes may further alter projected sea-level contributions. Given the importance of the grounding line region to our results, we expect further changes when considering three-dimensional shear fields, hydrology-ocean-bedrock-sediment interactions, iceberg calving, and uncertainties in predicted future surface mass balance. While these processes are unlikely to negate the effects we identify-given that effective pressure is largely isolated from external forcings-they underscore the complexity of grounding line

dynamics and the need for urgent model development to advance the fidelity of sea-level projections.

This study underscores the necessity of incorporating evolving hydrology systems in ice sheet models, showing that assumptions made when attempting to integrate realistic basal processes strongly influence sea-level rise projections. The interaction between ice dynamics and the hydrological system, as parameterised with HAF-scaling, leads to a much more mobile grounding line compared to a non-evolving hydrological state, even if produced with an hydrology model. While flotation at the grounding line with HAF-scaling is physically justified, the threshold used to determine the scaling area is a crude approximation that overlooks the complex role of the hydrological system, which influences ice dynamics based on the water volume it carries and thus varies between outlet glaciers and ice shelves.

## Methods
### Model setup
Our simulations do not capture a fully coupled ice-hydrology system, as such a model is not yet available for continental-scale applications. Instead, our simulations aim to determine whether a coupling is indeed necessary, rather than predetermining what might be considered a valid future projection of ice-mass change. We report mass loss estimates from drainage basins in both East and West Antarctica, showing how the parameterisation of the effective pressure exerts control on tipping points characterized by sharp increases in grounding line flux and rapid grounding line retreat in deep basins with retrograde bed slopes. The sea-level contributions in this study are computed based on changes in total ice volume above flotation by assuming zero or very small changes in the bedrock elevation due to isostatic adjustment and a constant ocean area of typically $3.625 \times 10^{14}$ m[251].

All experiments are conducted using the Shallow Shelf Approximation (SSA[52]) implementation of Elmer/Ice[53], an open-source finite-element ice dynamic model, to solve the vertically integrated ice dynamics. Upper and lower ice surface elevation and bedrock elevation are from the MEaSUREs BedMachine Antarctica, Version 3[54]. An unstructured finite-element mesh is created initially using Gmsh[55] for the Antarctic domain following coastline positions provided by BedMachine V3, and refined using MMG[56] based on ice thickness gradients and the Hessian matrix of the observed velocity fields provided by MEaSUREs Phase-Based Antarctica Ice Velocity map (Version 1)[57]. The low-computation cost of SSA model allows a high-resolution mesh of 1 km in fast-flowing regions and near the grounding line, with a coarser resolution (25 km) further inland. This represents the highest resolution used in continental Antarctic ice sheet models from ISMIP6 studies[2,5]. Sensitivity analysis using the LW sliding relation (Fig. S7) shows consistency between model outputs generated with 1 km mesh and 500 m mesh. Given the projected ice-mass loss using the Weertman sliding relation is more grid-size sensitive than the Coulomb relation used here[28,58], the mesh resolution artefact in this study is acceptable.

An inverse method is used to simulate both the basal drag coefficient $\beta$ and enhancement factor $\eta$ simultaneously through reducing the mismatch between the simulated and observed velocity. The inversion is performed with respect to a linear sliding coefficient:

$$\mathbf{\tau_b} = C_w \mathbf{u_b} = 10^{\beta} \mathbf{u_b} \tag{1}$$

where $u_b$ is the basal sliding velocity, $\beta$ is the sliding coefficient. The viscosity is defined by:

$$\eta = E_{\eta}^2 \eta_0 \tag{2}$$

where $\eta_0$ is a reference viscosity field calculated based on a prescribed and vertically integrated 3-D volume temperature distribution[59]. Three regularisation terms are introduced in the cost function to improve the conditioning of the inverse problem and balance between fitting with

observations and smoothness in $\beta$ and $\eta$. An 'L-surface' analysis[58] is conducted to determine the optimal regularisation parameters ($\lambda_{\beta} = 1e5$, $\lambda_{E_{\eta}1} = 1e6$ and $\lambda_{E_{\eta}2} = 0.02$). With the optimal parameters, we simulate the basal drag (Fig. S3a) and ice viscosity (Fig. S3b). The inverted basal drag is largely influenced by the bed geometry through affecting the ice flow resistance. The RMSEs between the simulated initial state in 1995 and observed ice thickness and ice surface velocity are shown in Fig. S8, which are among the low range of the RMSEs for the participating ice-sheet models in ISMIP6-2300 projections[5], except for the RMSE of velocity from RC_iN and RC_iN_HAF.

Taking the initial geometry and inverted basal drag and viscosity as the starting point at 1995, we conduct a historical run from 1995 to 2014 to relax the model as an initial adjustment. A projection run from 2015 to 2300 is followed with climate forcing derived from selected climate models under the ISMIP6-2300 protocol[60]. In the transient simulations, the surface mass balance (SMB) is given by the combination of a reference SMB and SMB anomalies, in which the reference SMB is the average mean SMB over 1995–2014 from the MAR product[61]. The basal melt rate underneath the ice shelf is parameterised following the simplified ISMIP6 standard basal melt parameterisation based on a prescribed relation between ocean thermal forcing and ice shelf melting rates[2]. Both SMB anomalies and thermal forcing are provided through the forcing provided by the ISMIP6 community[5]. We adopt the forcing using NorESM1-M/RCP8.5 model for the historical run and UKESM/ssp5-85 model for the projection run. Forcing data from other four climate models under high emission scenario (CCSM4/RCP8.5, HadGEM2/RCP8.5, CESM2/ssp5-85, NorESM1-M/RCP8.5) and two model under low emission scenario (UKESM/ssp1-26, NorESM1-M/RCP2.6) are applied in the projection run to explore the sensitivity to various climate forcing. For partially floating elements, the representation of basal friction follows the sub-element parameterisation 3 (SEP3) with 20 integration points per element[60], while the basal melt only is applied on fully floating cells recommended by ref. 28,58. The ice front is fixed with a minimum ice thickness of 40 m applied, which may result in a vast and thin future Ice Shelf and could potentially lead to an overestimated buttressing effect on the grounded ice, though the effect might be small.

### Experiment design
Two different sliding relations, linear Weertman and RC, are chosen in the transient simulations. For the RC relation, the friction coefficients are converted based on the inverted basal shear stress, $\mathbf{\tau_b} = 10^{\beta} \mathbf{u_b}$, to ensure that all experiments begin with the same initial basal shear stress (see details below). Importantly, the diverging transient behaviours arise from differences in how sliding evolves under the respective assumptions. To address uncertainties associated with the effective pressure, two formats of the RC sliding equations are considered (Table 1): one with explicit representation of effective pressure (hereafter 'RC_eN') and another with implicit representation of effective pressure through a spatially tuned parameter (hereafter 'RC_iN'). The transition between the Weertman and Coulomb-plastic behaviours differs in each format, as detailed below. Notably, in the Weertman regime, basal sliding is independent of effective pressure, making the transition critical for determining the extent of areas sensitive to spatial and temporal variations in effective pressure.

### Implicit representation of effective pressure
• Experiment RC_iN and RC_iN_HAF

Here we follow ref. 21's version of RC sliding relation and subsumed the roles of effective pressure,$N$, into the friction coefficient $C_1$:

$$\mathbf{\tau_b} = C_1 \left( \frac{\mathbf{u_b}}{\mathbf{u_b} + u_0} \right)^{\frac{1}{m}} \tag{3}$$

where $m$ is a flow exponent (set to 3). $C_1$ is converted based on Eq (1) and Eq (3):

$$C_1 = 10^\beta \mathbf{u_b} \left( \frac{\mathbf{u_b}}{\mathbf{u_b} + u_0} \right)^{-\frac{1}{m}} \quad (4)$$

Here we use $u_0 = 300$ m yr$^{-1}$ as the velocity threshold above which a Coulomb-plastic behaviour will occur. The model performance is proved to be relatively insensitive to $u_0$ values[21,49]. The Coulomb limit occurs when:

$$\mathbf{u_b} \gg u_0 \quad (5)$$

Then Eq (3) reduces to:

$$\tau_b \approx C_1 \quad (6)$$

In RC_iN, we assume $N$, and therefore $C_1$ obtained from inversion, do not change with time. However, this assumption may hold well for the interior of ice sheet but will break down in the fast flowing regions where the ice approaches flotation. To take this effect into account, a reduction on the basal shear stress $\tau_b$ is applied in experiment RC_iN_HAF so that $\tau_b$ can decrease smoothly towards zero at the grounding line, as shown below:

$$\mathbf{\tau_b} = \lambda C_2 \left( \frac{\mathbf{u_b}}{\mathbf{u_b} + u_0} \right)^{\frac{1}{m}} \quad (7)$$

$$\lambda = \begin{cases} 1 & \text{if } h_{af} \geq h_T \\ \frac{h_{af}}{h_T} & \text{if } h_{af} < h_T \end{cases} \quad (8)$$

where $h_{af}$ is the height-above flotation, $\lambda$ is the HAF-scaling factor, $h_T$ is the threshold. In such case, the basal shear stress near the grounding line will evolve over time with ice geometry. To choose a reasonable value of $h_T$, we calculate $\lambda C_2$ for each element in our model:

$$\lambda C_2 = \mathbf{\tau_b} \left( \frac{\mathbf{u_b}}{\mathbf{u_b} + u_0} \right)^{-\frac{1}{m}} \quad (9)$$

where $\tau_b$ and $u_b$ are simulated from inversion. The correlation between $\lambda C_2$ and $h_{af}$ (Fig. S9) indicates that $\lambda C_2$ reaches a relatively constant value of 0.21 Mpa where $h_{af} = 75$ m. When $h_{af} < 75$ m, a roughly linear trend is found, which indicates the linear decrease in basal drag when $h_{af}$ approaches zero. So we set $h_T$ to 75 m here and a sensitivity analysis on its values is discussed. In such case, $C_2$ is converted based on Eq (1) and Eq (8):

$$C_2 = 10^\beta \mathbf{u_b} \left( \frac{\mathbf{u_b}}{\mathbf{u_b} + u_0} \right)^{-\frac{1}{m}} \lambda^{-1} \quad (10)$$

## Explicit representation of effective pressure

The implemented Coulomb sliding relation can be written as[19]:

$$\mathbf{\tau_b} = C_3 N \left( \frac{\chi \mathbf{u_b}^{-m}}{1 + a\chi^q} \right)^{\frac{1}{m}} \mathbf{u_b} \quad (11)$$

$$a = \frac{(q-1)^{q-1}}{q^q} \quad (12)$$

$$\chi = \frac{\mathbf{u_b}}{C_3^m N^m A_s} \quad (13)$$

$q$ is a 'post-peak' exponent (set to 1), $C_3$ and $A_s$ are two different friction coefficients that affect the Coulomb and Weertman behaviour correspondingly. This equation can be rearranged to:

$$\mathbf{\tau_b} = C_3 N \left( \frac{\mathbf{u_b}}{\mathbf{u_b} + C_3^m N^m A_s} \right)^{\frac{1}{m}} \quad (14)$$

The Coulomb limit occurs when the basal sliding resistance $\tau_b$ is primarily determined by $C_3 N$, with minimal influence from the basal sliding velocity. This happens in regions where the sliding relatin transitions from Weertman regime to Coulomb regime when:

$$\mathbf{u_b} \gg C_3^m N^m A_s \quad (15)$$

Under this condition, Eq (14) reduces to:

$$\tau_b \approx C_3 N \quad (16)$$

This is classic Coulomb friction law, where basal shear stress $\tau_b$ is directly proportional to the effective pressure $N$ and the Coulomb coefficient $C_3$. We now need to convert the linear Weertman coefficient $C_W = 10^\beta$ to the unknown Coulomb coefficients $C_3$ and $A_s$. The sliding parameter $A_s$ determines the ease of basal sliding, with higher values representing more resistance to sliding, which is influenced by both physical properties of the ice-bed interface and the effective pressure[18]. Here we interpret the inverted linear friction coefficient $\beta$ everywhere in terms of a non-linear Weertman coefficient, using the following relation:

$$A_s = |\mathbf{u_b}|^{1-m} 10^{-m\beta} \tanh(2N) \quad (17)$$

Eq (14) is rearranged in terms of the Coulomb $C$ parameter:

$$C_3 = |\mathbf{u_b}| 10^\beta N^{-1} (1 - 10^{m\beta} |\mathbf{u_b}|^{m-1} A_s)^{-1/m} \quad (18)$$

- Experiment RC_eN_POC

In this experiment, we assume a perfect hydrological connection between the subglacial drainage system and the ocean so that $N$ can be calculated based on ice thickness and bedrock elevation:

$$N = \rho_i g H + \rho_o g z_b \quad (19)$$

where $H$ is ice thickness, $g$ is acceleration due to gravity, $\rho_i$ is ice density, $\rho_o$ is ocean water density, $z_b$ is bedrock elevation. It allows $N$ to evolve with ice geometry under the assumption that subglacial water pressure equals the ocean pressure at the grounding line, which is more realistic near the grounding line when ice approaches flotation.

- Experiments RC_eN_GlaDS and RC_eN_GlaDS_HAF

In experiments RC_eN_GlaDS and RC_eN_GlaDS_HAF, the basal drag is decided following Eq. (14) and $N$ is simulated with the Glacier Drainage System model (GlaDS[37]), implemented in Elmer/Ice as a separate module[62]. The GlaDS model was selected for this study as it combines a distributed water sheet and channelized drainage system, providing a good representation of subglacial hydrology that is particularly suited to investigating how bed topography influences effective pressure and basal drag[9,35,37,62]. It is fully implemented within Elmer/Ice, allowing for seamless spin-up of the hydrology model with a restart from the ice sheet model. GlaDS is run into a steady state using the initial ice geometry to calculate the spatial distribution of effective pressure at the start of the simulation[63]. In this setup, frictional heating (calculated using the inversion simulations described above) is assumed to be the sole source of subglacial meltwater production, as it

dominates in areas of significant basal sliding, where high ice velocities generate far more heat than the relatively small and more uniform geothermal flux[64]. To ensure that the hydrology system reaches a steady state, the total subglacial water budget is held constant throughout the spin-up period (over ~100 model years). This assumption introduces negligible variability in the final results compared to alternative assumptions on subglacial melt supply.

Bedrock geometry and ice thickness strongly influences subglacial water flow and channel distribution, as lower-elevation areas, such as valleys, accumulate water due to hydraulic potential gradients shaped by bedrock topography and ice thickness, resulting in higher water pressures and lower effective pressures. In RC_eN_GlaDS, we assume the initially computed effective pressure to remain constant with time. To account for a temporally evolving effective pressure near the grounding line, the previously described HAF-scaling is applied on effective pressure in RC_eN_GlaDS_HAF when $h_{af} < h_T$, defining it as $N^*$.

$$\mathbf{\tau_b} = C_4 N^* \left( \frac{\mathbf{u_b}}{\mathbf{u_b} + C_3^m N^{*m} A_s} \right)^{\frac{1}{m}} \quad (20)$$

$$N^* = \lambda N \quad (21)$$

$$\lambda = \begin{cases} 1 & \text{if } h_{af} \geq h_T \\ \frac{h_{af}}{h_T} & \text{if } h_{af} < h_T \end{cases} \quad (22)$$

$$A_s = |\mathbf{u_b}|^{1-m} 10^{-m\beta} \tanh(2 \cdot N^*) \quad (23)$$

$C_4$ is converted based on Eq (1):

$$C_4 = |\mathbf{u_b}| 10^{\beta} N^{*-1} (1 - 10^{m\beta} |\mathbf{u_b}|^{m-1} A_s)^{-1/m} \quad (24)$$

## Data availability
Original forcings datasets from various climate models are available on Ghub https://theghub.org/dataset-listing under 'ISMIP6 Antarctica 2300'. The MEaSUREs BedMachine Antarctica, Version 3 is available at https://nsidc.org/data/nsidc-0756/versions/3. The MEaSUREs Phase-Based Antarctica Ice Velocity map (Version 1) is available at https://nsidc.org/data/nsidc-0754/versions/1.The model configurations and all data and scripts used to generate the figures are available at https://doi.org/10.5281/zenodo.14874036.

## Code availability
The most recent developments of the Elmer/Ice code are downloadable from https://github.com/ElmerCSC/elmerfem (last visited 2024-08-10). The most recent release of Elmer/Ice is available from https://doi.org/10.5281/zenodo.7892181.

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

## Acknowledgements

C.Z. is the recipient of an Australian Research Council Discovery Early Career Researcher Award (project number DE240100267) funded by the Australian Government. C.Z., Y.W., P.C., and B.K.G. received grant funding from the Australian Government as part of the Antarctic Science Collaboration Initiative program (ASCI000002). R.G. was supported by the Research Council of Finland (grant numbers 322430 and 355572), and by the Finnish Ministry of Education and Culture and CSC-IT Center for Science (Decision diary number OKM/10/524/2022). T.Z. was supported by the Research Council of Finland (grant number 322978). E.J. was supported by the Research Council of Finland (grant number 355783). P.C., B.K.G. and M.A.K. are supported by the Australian Research Council Special Research Initiative, Australian Centre for

Excellence in Antarctic Science (project number SR200100008). J.C. and F.G.-C. received funding from the French National Research Agency (ANR) under grants ANR-19-CE01-0015 (EIS). P.M. received funding from Agence Nationale de la Recherche—France 2030 as part of the PEPR TRACCS programme under grant number ANR-22-EXTR-0010. All simulations were enabled by computational resources provided by CSC-IT Center for Science Ltd.

## Author contributions

C.Z. led the research, conducted the simulations, analysed the results, led the writing. R.G. conducted the GlaDS simulation. R.G. and T.Z. contributed to the experiment design. R.G., T.Z., F.G.-C., Y.W., J.C., P.M., L.S. and E.J. contributed to the model setup and analysis. R.G., T.Z., F.G., Y.W., J.C., P.M., L.S., E.J., B.K.G., P.C. and M.A.K. contributed to the manuscript writing.

## Competing interests

The authors declare no competing interests.
