## [Peer Review file · Nature Communications]

Subglacial Water Amplifies Antarctic Contributions to Sea-Level Rise

Corresponding Author: Dr Chen Zhao

Version 0:

Reviewer comments:

Reviewer #1

(Remarks to the Author)

1. What are the element sizes in the Model and how dependent is the SLE for the different scenarios on the mesh size. In general you should in a few lines give basic information on model parameters
2. Is there a correlation between the bed topography and the distribution of effective pressures and inverted basal drag. If so you should point it out
3. I would like to point to a publication where although in a different manner the same problem is addressed and maybe you can use it to further complete your argument.
Goeller et al.: A balanced water layer concept for subglacial hydrology in large-scale ice sheet models, *The Cryosphere*, 7, 1095-1106, 2013 doi:10.5194/tc-7-1095-2013
4. Finally - from a person who is color blind - you would make life much easier for about 10 % of your readers if you would use different line styles in addition to colors

Reviewer #2

(Remarks to the Author)

Subglacial Water Amplifies Antarctic Contributions to Sea-Level Rise

The present manuscript investigates the effect of different variants for computing the basal pressure below the Antarctic ice sheet used in the basal sliding law. Basal sliding is a key component in ice sheet modeling that is hard to constrain due to a lack of direct observations and it has a strong influence on the predicted sea level rise and timing of tipping points. Using the ice sheet model Elmer/Ice, the authors conduct a number of different experiments each with a different treatment for computing the effective pressure N (ice overburden pressure minus subglacial water pressure) and investigate the effect under several climate forcing scenarios. They show that different Antarctic sectors react differently to the sliding relation chosen, depending mostly on the topography and dominating driving stress regime. This study is original in that it focuses on comparing the effects of different sliding relations within the same ice sheet model. Subglacial hydrology is an important missing piece in ice sheet modeling (and especially relevant for sea level rise predictions), and while the paper does not examine the validity of them, it investigates the implications of the different approaches that are used in the wild.

REMARKS

Discussion of sliding laws

In the introduction, the authors compare the two overarching classes of sliding laws: classical Weertman and regularized Coulomb and state that models often ignore or crudely implement the effective pressure N "by adopting the Weertman

relation as the commonplace approach” (L. 39f). While this is generally correct, I would prefer a more nuanced statement because the current formulation can imply that most models have no treatment of effective pressure at all. From skimming over Seroussi et al. (2020), I noticed that out of the 13 used models, only 4 to 6 seem to use classical Weertman without considering basal pressure (unfortunately, not all of them clearly state their sliding relation). Maybe the variety in the approaches can be mentioned in the paper. The following paragraph on HAF (L. 45ff) describes one simple approach for including N, but I think it could be better integrated with the previous part.

Sliding laws used in ISMIP6:

AWI_PISM: Schoof/Coulomb

DOE_MALI: linear basal friction law with spatially varying basal friction coefficient

ILTS_PIK_SICOPOLIS: Weertman–Budd-type sliding law with sub-melt sliding (Sato and Greve, 2012) and subglacial hydrology IMAU_IMAUICE: ?? probably Schoof/Coulomb? citing Bueler Brown 2009 JPL_ISSM: ??? probably coulomb-like, citing Morlighem et al. 2010

LCSE_GRISLI: power law basal friction (Weertman)

NCAR_CISM: basal sliding is similar to that of Schoof (2005), combining power law and Coulomb behavior PIK_PISM:

Mohr-Coulomb UCIJPL_ISSM: ??? ULB_FETISH: power law basal sliding UTAS_ELMERICE: linear sliding relation VUB_AISMPALEO: Weertman, but with height above bouancy VUW_PISM: probably Schoof/Coulomb

2. Explaining the experiments in the main text

The Method section does a great job of explaining the different experiments, but it would be helpful for the reader to get a short overview already in the main text to facilitate the understanding of the results. It would be enough to briefly explain the different treatments of N and possibly further explain the difference between HAF and POC (they seem similar in that they both have the goal of enforcing floating condition at the grounding line).

I would also appreciate a comment on why these particular treatments were chosen; are they representative of what is used in current models?

It would be good to lead the reader along from simple implementations to more complex/realistic and explain why you chose the GlaDS model out of a variety of subglacial hydrology models (see e.g. De Fleurian et al. 2018).

I am a little bit concerned with the validity of the inversion result for future projections when the basal conditions (especially at the grounding line) have changed. This affects mostly the experiments where N is treated implicitly (as part of the inversion). Are they still a good baseline after e.g. a substantial retreat of the GL which would imply a major reconfiguration of the hydraulic system?

3. Structure of results and headlines

I believe the presentation of the results could be enhanced for better readability and comprehension. While the beginning that focuses on the implicit handling of N does a good job of describing the difference between LW and RC_iN_HAF, the following parts could be arranged in a more organized way, as I experienced them hard to follow. One idea could be to adapt something like the following structure (very similar to the current one):

Keep the first part on implicit handling of N: LW vs RC_iN_HAF as it is (L. 59 to L. 80).

Then have a second part that covers the explicit handling of N: RC_eN_GlaDS vs RC_eN_POC vs RC_eN_GlaDS_HAF. Here I would suggest to start with RC_eN_POC as the most simple approach and then introduce GlaDS and the extended GlaDS (RC_eN_GlaDS_HAF).

With this continental view established, continue with the Tipping points and the regional analysis.

Not much change is needed, but these sections should be clearly identifiable and might deserve their own headings, also because the current in-between headings (“Subglacial effective pressure must be included in sliding relations” and “Subglacial effective pressure controls ice fluxes”) are very general and do not help much to navigate the text.

The final part before the discussion (“Increased sensitivity to effective pressure near the grounding line”) is in a good place.

At some point, Kazmierczak et al. (2022) should also be referenced, as they also investigate the influence of different sliding relations for Antarctica.

Remarks by lines

L. 29: also frictional heating due to internal deformation of ice

L. 45f: Can't the effect of N be also approximated in the sliding coefficients for Weertman sliding laws? see also concern 1.

L. 48: more realistic than what?

L. 56: Why SSA, what is the implication?

L. 57: Why is SSA used here? Why not hybrid/higher order? What are the implications?

L. 60: Why is RC_iN_HAF more realistic? is combined effects of seawater intrusion and meltwater production is also considered in the eN variants? is this only about RC vs LW? more explanation is needed.

L. 68: Replace “near mass balance” with “near equilibrium”. I don't understand the second part of the sentence regarding

observed mass loss.

L.71: "For WAIS...": Sentence is unclear. Maybe "For the WAIS the experiment using the RC relation and high emission scenario,..."

L. 73: produce.

L. 74: "However, the RC..." -> "However, using the RC..."

L. 75: "than LW" -> "than using the LW relation"
The whole paragraph seems to be missing words.

L. 82: This sentence is not very clear, seems obvious.

L 99ff: Isn't the addition of HAF to the GlaDS N a little bit of double counting, because the effect of the ocean is already included in the GlaDS model? Though it is only appropriate at the time where N was computed in GlaDS.

L. 106: Explain coulomb limit

L. 115-122: Can the experiments using POC produce meaningful results for basins with reverse bed slope? Won't POC on the retrograde bed will lead to higher basal pressure in the inland than at the grounding line?

L. 153: How valid is the distribution of N computed by GlaDS for present day conditions after more than a century? Even with the HAF scaling, the whole hydraulic regime might have shifted.

L. 163: What does "constrained future surface mass budgets estimates" mean? Improvement of the predictions of SMB?

Figure 1:
good colors, bigger fonts, no box around legend

Figure 2:
caption: "sea-level contributions between linear Weertman law" -> "sea-level contributions in experiment using linear Weertman law".

Figure 4:
caption: "with a Hydrology model" -> "from GlaDS"
Does channel area mean cross section of individual channels?
why is a diverging colormap used? What is special about 0.5 MPa?
small inset figures in G-H and D-Dp are hardly readable.

Figure 5:
Why are the lines dashed?

References

DE FLEURIAN B, WERDER MA, BEYER S, et al. SHMIP The subglacial hydrology model intercomparison Project. *Journal of Glaciology*. 2018;64(248):897-916. doi:10.1017/jog.2018.78

Kazmierczak, E., Sun, S., Coulon, V., and Pattyn, F.: Subglacial hydrology modulates basal sliding response of the Antarctic ice sheet to climate forcing, *The Cryosphere*, 16, 4537–4552, <https://doi.org/10.5194/tc-16-4537-2022>, 2022.

Bueler, E., and J. Brown (2009), Shallow shelf approximation as a "sliding law" in a thermomechanically coupled ice sheet model, *J. Geophys. Res.*, 114, F03008, doi:10.1029/2008JF001179.

Morlighem, M., E. Rignot, H. Seroussi, E. Larour, H. Ben Dhia, and D. Aubry (2010), Spatial patterns of basal drag inferred using control methods from a full-Stokes and simpler models for Pine Island Glacier, West Antarctica, *Geophys. Res. Lett.*, 37, L14502, doi:10.1029/2010GL043853.

Reviewer #3

(Remarks to the Author)

Review of "Subglacial Water Amplifies Antarctic Contributions to Sea-Level Rise", for *Nature Communications*, by Zhao, Gladstone, Zwinger, Gillet-Chaulet, Wang, Caillet, Mathiot, Saraste, Jager, Galton-Fenzi, Christoffersen and King

This manuscript describes the importance of incorporating subglacial hydrology parameterizations into whole Antarctic ice sheet models, by varying parameterization of basal water effective pressure in a basal slip relation. The authors conclude

that effective pressure is required in the slip relationship, and that ice flux is sensitive to effective pressure model, especially near the grounding line.

We believe the topic of this manuscript is important and can be of interest to a wide range of readers. However, we believe that this manuscript can be significantly improved by considering the comments listed below.

For now, the manuscript is “expert-facing”, i.e., it is assumed that the readers have a substantial understanding of the background and methods of this study. The manuscript leaps over many (essential) details that could have been helpful for the readers to follow along.

- For an important study that can be of interest to the general public, especially in the journal of Nature Communication, we argue this is not appropriate. We recommend the authors to provide more detailed elaborations on the background, methods, findings, and implications of this study. A few examples can be found in the comments below.
- In the section “Subglacial effective pressure must be included in sliding relations”; it implied the observed mass gain of the linear Weertman models demonstrate the need for subglacial effective pressure. The authors need to make a stronger case - ie demonstrate that mass gain is not plausible over these time frames.
- We recommend the authors to elaborate on the differences between model setups. We appreciate the balance between the amount of work presented in this manuscript and the space limitations. But the current manuscript makes it really difficult to follow the different model setups, especially considering the number of runs that are being presented and discussed.
- It is not clear why/how including different subglacial hydrology components can change the AIS SLR projection by such a large degree. What is the physical process behind the differences? We recommend the authors to elaborate on this and make it more clear.
- Line 47-52: HAF is a very important concept for this study. However, the authors didn't provide any explanation/elaboration on the concept, other than providing one reference. (This is also true for some other important concepts throughout the manuscript.) This approach makes it difficult for non-expert readers to understand the work and follow the logic flow. We recommend providing a more detailed explanation of the concept, and/or maybe even a conceptual diagram.
- It is not clear what assumption is made about the subglacial meltwater budget. Does the model run only require a certain and finite amount of subglacial water? Or do the authors assume an “unlimited” supply of subglacial water generated by basal melting upstream? And does a limited subglacial meltwater budget change the model output?
- * The authors need to address the limitation of the study in the Discussion section. We understand that the authors need to consider the space limitations per journal guideline, but we strongly recommend the authors to reconsider the balance between (1) further elaboration and discussion of the study findings, (2) connection with other studies, (3) broader implications, and (4) limitation of the study, for the Discussion section.

minor comments:

* This manuscript uses a lot of acronyms - are they really necessary? Like, “GL” for “grounding line”? Having too many acronyms can be counterproductive. Also, it may help to spell out the run names in Figure 1.

* We noticed that no doi/web-link is provided for the references. We encourage the authors to consider including such information.

Unsure of what is meant by the second and third authors contributed equally.

Reviewer #4

(Remarks to the Author)

Version 1:

Reviewer comments:

Reviewer #1

(Remarks to the Author)

The paper is the result of a numerical study investigating the effect of basal pressure under the Antarctic ice sheet and the role of the pressurized water layer in basal slip. While for a direct prognostic application there is lack of data, the paper defines the range of solutions for Ice Sheet/iceshelf dynamics depending on pressure in particular at the icesheet-iceshelf boundary.

The authors have responded to all reviewers comments and i am satisfied with the answers to my specific comments.

Reviewer #2

(Remarks to the Author)

The authors have addressed the reviewers' concerns effectively. I'm pleased with the improved paper structure and extended description. I have only one minor comment:

I appreciate the more detailed description of sliding laws used in the ISMIP/ABUMIP experiments, but I would like to point out that the PISM model uses a Mohr-Coulomb type sliding law and not a Weertman type. It is specifically mentioned in the manual that Weertman should not be used with PISM: <https://www.pism.io/docs/manual/modeling-choices/dynamics/weertman.html> and also in the model description it is pretty clear:

Seroussi et al. 2020

PIK_PISM

We apply a power law for sliding with a Mohr-Coulomb criterion relating the yield stress to parameterized till material properties and the effective pressure of the overlying ice on the saturated till (Bueler and van Pelt, 2015).

Seroussi et al. 2024:

PIK_PISM

A generalized power law (Schoof & Hindmarsh, 2010) is applied to parameterize basal sliding. The basal friction coefficient depends on the effective pressure and till friction angle, that is parameterized using a heuristic, piecewise linear function of the bed elevation (Martin et al., 2011).

I know from personal communication that AWI_PISM uses the same setup in that regard.

This does of course not change the overall conclusion of the authors that Weertman type sliding laws are dominant in the benchmarks.

Reviewer #3

(Remarks to the Author)

This work illustrates the need for improved knowledge of the evolution of subglacial hydrology, and we recommend publication.

The revised manuscript reads much better with acronym expansion and with more detailed concept explanations and discussions. Our only minor follow-up suggestion is to explicitly state in the Method section that the subglacial water budget is held constant because (1) the model requires such assumption to reach steady state, and (2) different assumptions on this do not cause significant variation in the modeling results.

Reviewer #4

(Remarks to the Author)

Response to Reviewers

Many thanks to the reviewers for their valuable comments. Our responses are highlighted in blue, and the line numbers refer to the revised manuscript.

Reviewer #1 (Remarks to the Author):

1. What is the element sizes in the Model and how dependent is the SLE for the different scenarios on the mesh size. In general you should in a few lines give basic information on model parameters.

Thanks for the comments. We have included the element size on the *Method Setup* Section in the original manuscript **“The low-computation cost of SSA model allows a high-resolution mesh of 1 km in fast-flowing regions and near the GL and a coarser resolution (25 km) further inland.”**

We have done sensitivity tests to different mesh resolutions (500 m, 1 km, 2 km) near the grounding line with the linear Weertman relation (**LW**). See the figure below. It suggests that higher resolution would predict less mass loss than coarser resolution, which is consistent with similar sensitivity tests done on Wilkes Subglacial Basin (*Wang et al. 2024*) and an idealised domain (*Seroussi et al., 2018*). It can be seen that the 1 km mesh used in this study suggests a more consistent result with 500 m mesh compared with the 2 km mesh. Previous studies (*Wang et al., 2024; Seroussi et al., 2018*) indicate that the Regularised Coulomb sliding relation with effective pressure included is less sensitive to the mesh resolution compared with **LW**. So we think the sensitivity of sea-level rise projections to the mesh size is little compared with the dependence on the different treatment of effective pressure.

To make it clearer and include the information above, we added the figure below into the supplementary material and a few sentences in the *Methods* section (Line 426-431).

“The low-computation cost of SSA model allows a high-resolution mesh of 1 km in fast-flowing regions and near the grounding line, with a coarser resolution (25 km) further inland. This represents the highest resolution used in continental Antarctic ice sheet models from ISMIP6 studies^{8,9}. Sensitivity analysis using the LW sliding relation (Fig. S7) shows consistency between model outputs generated with 1 km and 500 m meshes. Given the projected ice mass loss using the Weertman sliding relation is more grid-size sensitive than the Coulomb relation used here^{10,11}, the mesh resolution artefact in this study is acceptable.”

Figure S7. Sensitivity of sea-level contributions with linear Weertman law to various mesh resolutions.

2. is there a correlation between the bed topography and the distribution of effective pressures und inverted basal drag. If so you should point it out

Thank you for your comment. There is indeed a correlation between the bed topography and the distribution of effective pressure and inverted basal drag. Low-lying areas tend to have lower effective pressure and basal drag, while elevated regions exhibit higher values. To include this, we add one sentence in the *Methods* section (Line 520-523):

“Bedrock geometry and ice thickness strongly influences subglacial water flow and channel distribution, as lower-elevation areas, such as valleys, accumulate water due to hydraulic potential gradients shaped by bedrock topography and ice thickness, resulting in higher water pressures and lower effective pressures.”

The correlation between bed topography and inverted basal drag is mediated by how bed geometry influences ice flow resistance, with high basal drag generally associated with elevated or rough areas and low basal drag linked to smooth, low-lying regions. To include this, we add one sentence in the inversion section (Line 440-441):

“The inverted basal drag is largely influenced by the bed geometry through affecting the ice flow resistance.”

3. I would like to point to a publication where although in a different manner the same problem is addressed and maybe you can use it to further complete your argument.

Goeller et al.:A balanced water layer concept for subglacial hydrology in large-scale ice sheet models, *The Cryosphere*, 7, 1095-1106,2013 doi:10.5194/tc-7-1095-2013

Many thanks for recommending this valuable paper. This paper introduces the balanced water layer concept, which provides a simplified representation of subglacial hydrology suitable for large-scale ice sheet models. We have cited this paper in our *Introduction*:

“The widespread production of subglacial water beneath the Antarctic Ice Sheet, and the potential intrusion of warm seawater near the grounding line, plays a crucial role in modulating basal sliding and enabling rapid ice flow⁷⁻¹¹.” (Line 26-27)

4. Finally - from a person which is color blind - you would make life much easier for about 10 % of your readers if you would use different line styles ind addition to colors

Thank you for highlighting this important point. Although Reviewer 2 noted that Figure 1 uses good colors, we have updated all figures to ensure they are more color-blind-friendly by incorporating both improved color schemes and distinct line styles.

Reviewer #2 (Remarks to the Author):

Subglacial Water Amplifies Antarctic Contributions to Sea-Level Rise

The present manuscript investigates the effect of different variants for computing the basal pressure below the Antarctic ice sheet used in the basal sliding law. Basal sliding is a key component in ice sheet modeling that is hard to constrain due to a lack of direct observations and it has a strong influence on the predicted sea level rise and timing of tipping points.

Using the Ice sheet model Elmer/Ice, the authors conduct a number of different experiments each with a different treatment for computing the effective pressure N (ice overburden pressure minus subglacial water pressure) and investigate the effect under several climate forcing scenarios. They show that different Antarctic sectors react differently to the sliding relation chosen, depending mostly on the topography and dominating driving stress regime.

This study is original in that it focuses on comparing the effects of different sliding relations within the same ice sheet model. Subglacial hydrology is an important missing piece in ice sheet modeling (and especially relevant for sea level rise predictions), and while the paper does not examine the validity of them, it investigates the implications of the different approaches that are used in the wild.

REMARKS

Discussion of sliding laws

In the introduction, the authors compare the two overarching classes of sliding laws: classical Weertman and regularized Coulomb and state that models often ignore or crudely implement the effective pressure N “by adopting the Weertman relation as the commonplace approach” (L. 39f). While this is generally correct, I would prefer a more nuanced statement because the current formulation can imply that most models have no treatment of effective pressure at all. From skimming over Seroussi et al. (2020), I noticed that out of the 13 used models, only 4 to 6 seem to use classical Weertman without considering basal pressure (unfortunately, not all of them clearly state their sliding relation). Maybe the variety in the approaches can be mentioned in the paper.

Sliding laws used in ISMIP6:

AWI_PISM: Schoof/Coulomb

DOE_MALI: linear basal friction law with spatially varying basal friction coefficient
ILTS_PIK_SICOPOLIS: Weertman–Budd-type sliding law with sub-melt sliding (Sato and Greve, 2012) and subglacial hydrology^[1] IMAU_IMAUICE: ?? probably Schoof/Coulomb? citing Bueler Brown 2009^[1] JPL_ISSM: ??? probably coulomb-like, citing Morlighem et al. 2010

LCSE_GRISLI: power law basal friction (Weertman)

NCAR_CISM: basal sliding is similar to that of Schoof (2005), combining power law and Coulomb behavior^[1] PIK_PISM: Mohr-Coulomb^[1] UCIJPL_ISSM: ???^[1] ULB_FETISH: power law basal sliding^[1] UTAS_ELMERICE: linear sliding relation^[1] VUB_AISMPALEO: Weertman, but with height above bouancy^[1] VUW_PISM: probably Schoof/Coulomb

Thanks for the suggestions and providing the sliding law used in the ISMIP6-2100 projections. However, I would like to clarify that, based on Seroussi et al. 2020, 13 out of 16 models in ISMIP6-2100 have used Weerman law.

To clarify and respond to this comment we have included a table that summarises the sliding laws used in all recent model inter-comparison projects (ISMIP6-2100, ISMIP6-2300, and ABUMIP) in **Table S1** (Supplementary Material).

In Line 47-51, we added the text:

“Because the actual distribution of effective pressure under the Antarctic Ice Sheet is unknown, model-based sea-level rise projections typically either use Weertman sliding everywhere, which does not include effective pressure (e.g., refs.^{24–26}), or make simple assumptions about effective pressure when applying RC sliding, (e.g., refs.^{27–29}). A summary of sliding laws used in three major Model Intercomparison Projects reveals that the majority of Antarctic Ice Sheet models (e.g., 13 out of 16 models in ISMIP6-2100², 9 out of 16 models in ISMIP6-2300⁵, and 7 out of 15 models in ABUMIP³⁰) employed the Weertman sliding relation.”

ISMIP6-2100 ⁶		ISMIP6-2300 ⁷		ABUMIP ⁸	
Model Name	Sliding Law	Model Name	Sliding Law	Model Name	Sliding Law
AWI_PISM	Weertman(m=3)	-	-	AWI_PISMPal	Coulomb
-	-	UCM_Yelmo	Regularised Coulomb	-	-
IMAU_IMAUICE1	Weertman(m=3)	MAU_UFEMISM	Regularised Coulomb	IMAU_ICE	Coulomb
IMAU_IMAUICE2	Weertman(m=3)	-	-	-	-
NCAR_CISM	Weertman(m=3)-Coulomb	NCAR_CISM1	Weertman(m=3)-Coulomb	NCAR_CISM	Weertman(m=3)-Coulomb
-	-	NCAR_CISM2	Zoet-Iversen	-	-
-	-	NORCE_CISM	Zoet-Iversen	-	-
DOE_MALI	Weertman(m=1)	DOE_MALI	Weertman (m=3)	DOE_MALI	Weertman(m=1)
ILTS_PIK_SICOPOLIS	Weertman(m=3)-Budd	ILTS_PIK_SICOPOLIS	Weertman(m=3)-Budd	ILTS_PIK_SICOPOLIS	Weertman(m=3)
JPL_ISSM	Budd(m=1)	UCSD_ISSM	Weertman(m=3)-Budd	JPL_ISSM	Weertman(m=1)
UCIPL_ISSM	Weertman(m=3)	-	-	-	-
PIK_PISM1	Weertman(m=3)	PIK_PISM	Weertman(m=3)	PIK_PISM	Coulomb
PIK_PISM2	Weertman(m=3)	-	-	-	-
ULB_FETISH_16km	Weertman(m=2)	ULB_FETISH	Weertman(m=3)	ULB_FETISH	Weertman(m=2)
ULB_FETISH_32km	Weertman(m=2)	-	-	-	-
UTAS_ELMERICE	Weertman(m=1)	UTAS_ELMERICE	Weertman (m=1)	-	-
-	-	IGE_ElmerIce	Weertman (m=1)	IGE_ElmerIce	Weertman (m=3)
VUB_AISMPALEO	Weertman(m=3)	-	-	-	-
VUW_PISM	Weertman(m=3)	VUW_PISM1	Weertman(m=3)	-	-
-	-	VUW_PISM2	Weertman(m=3)	-	-
-	-	-	-	ARC_PISM1	Coulomb
-	-	-	-	ARC_PISM2	Coulomb
LCSE_GRISLI	Weertman(m=3)	LCSE_GRISLI	Weertman(m=3)	LCSE_GRISLI	Coulomb
-	-	LCSE_GRISLI2	Weertman(m=3)	-	-
-	-	-	-	CPOM_BISICLES	Weertman(m=3)-Coulomb
-	-	-	-	PSU_PSU3D1	Weertman(m=2)
-	-	-	-	PSU_PSU3D2	Weertman(m=2)

Table S1. Sliding law used in ISMIP6-2100 Antarctic⁶, ISMIP6-2300 Antarctic⁷, and ABUMIP⁸.

The following paragraph on HAF (L. 45ff) describes one simple approach for including N, but I think it could be better integrated with the previous part.

To better illustrate the differences in how effective pressure is treated across experiments, we have reorganized the *Introduction* as suggested by Reviewer 2. We now introduce the explicit representation of effective pressure first (Line 52-59), followed by the implicit representation (Line 60-63). Additionally, we have added a separate paragraph (Line 64-73) specifically discussing HAF-scaling. To further clarify the experimental design, we have also moved **Table 1** from the *Methods* section to the *Introduction*.

2. Explaining the experiments in the main text

The Method section does a great job of explaining the different experiments, but it would be helpful for the reader to get a short overview already in the main text to facilitate the understanding of the results. It would be enough to briefly explain the different treatments of N and possibly further explain the difference between HAF and POC (they seem similar in that they both have the goal of enforcing floating condition at the grounding line).

Thanks for the helpful suggestions. As mentioned in our response to the comment above, we reorganized the *Introduction* to better explain the difference between experiments

and added a summary paragraph at the end (Line 74-81). A modified **Table 1** is moved to the *Introduction* to help better interpret the differences.

To explain the differences between HAF and POC, we added two sentences at Line 70-73:

“Both the HAF-scaling and perfect ocean connection assumption enforce the floating condition at the grounding line. However, HAF-scaling, applied only near the grounding line, adjusts basal resistance based on ice elevation relative to flotation, potentially under- or overestimating effective pressure. In contrast, the perfect ocean connection assumes uniform hydraulic connectivity across the domain, systematically overestimating effective pressure by neglecting hydraulic potential gradients.”

I would also appreciate a comment on why these particular treatments were chosen; are they representative of what is used in current models?

In the reorganized *Introduction* section, we now provide a clearer and more logical explanation for why these specific treatments were chosen, with appropriate citations as evidence for their use in current models. To be consistent with the logical structure of the *Results* section suggested by Reviewer 2, we begin with the explicit representation of effective pressure, employing widely recognized approaches: the *perfect ocean connection* assumption, which is commonly used to enforce the floating condition at the grounding line (Line 52-55), and an alternative approach that derives effective pressure using a subglacial hydrology model (GlaDS) to account for spatial variability (Line 56-59). We then include the implicit representation of effective pressure, as used in Pine Island Glacier simulations by Joughin et al. (2019). This approach, chosen for its computational efficiency and ability to reproduce observed ice flow speeds through spatially optimized parameters, remains a practical and widely adopted method for large-scale ice sheet simulations (Line 60-63). Then a separate paragraph (Line 64-73) about Height Above Flotation scaling (HAF-scaling) is added to manually scale the basal resistance or the effective pressure from a prescribed HAF value down to zero at the grounding line. The last paragraph is a summary of experiment design in this study with a modified **Table 1**.

It would be good to lead the reader along from simple implementations to more complex/realistic and explain why you chose the GlaDS model out of a variety of subglacial hydrology models (see e.g. De Fleurian et al. 2018).

Thank you for your comment. We use the GlaDS model because it was designed for glacial applications (Werder et al., 2013) and has been successfully utilized in previous ice sheet studies (Gagliardini et al., 2018, Dow et al. 2022, Pelle et al. 2024, etc). Introducing an alternative model would be impractical and time-consuming.

GlaDS is in fact particularly well-suited for our study as it represents a continuum approach that combines distributed and channelized drainage systems, making it ideal for capturing the diverse hydrological processes that occur beneath glaciers and ice sheets. Its ability to simulate both water-filled channels and sheet-like flows aligns well with the goals of our research, particularly in exploring how bed topography influences effective pressure and basal drag.

While other hydrology models exist, such as those focused exclusively on either distributed or channelized systems, GlaDS offers a balance of physical realism and computational feasibility, making it a robust choice for studying subglacial hydrology in coupled ice sheet modeling frameworks. This combination of practical implementation within Elmer/Ice and its theoretical strengths guided our decision to use GlaDS in this study.

To address this comment in the manuscript, we include a brief explanation in the *Methods* section (Line 511-515):

"The GlaDS model was selected for this study as it combines a distributed water sheet and channelized drainage system, providing a good representation of subglacial hydrology that is particularly suited to investigating how bed topography influences effective pressure and basal drag¹⁹⁻²². It is fully implemented within Elmer/Ice, allowing for seamless spin-up of the hydrology model with a restart from the ice sheet model."

I am a little bit concerned with the validity of the inversion result for future projections when the basal conditions (especially at the grounding line) have changed. This affects mostly the experiments where N is treated implicitly (as part of the inversion). Are they still a good baseline after e.g. a substantial retreat of the GL which would imply a major reconfiguration of the hydraulic system?

We share the reviewer's concern, which is precisely what this study aims to emphasize. We agree that basal drag should vary over time, especially near the grounding line.

As the reviewer pointed out, in experiment with an implicit effective pressure (RC_iN), assuming effective pressure into a constant parameter fails to account for the temporal variability of effective pressure in fast-flowing regions, which is why we use HAF-scaling to manually reduce basal drag near the grounding line in RC_iN_HAF. Similarly, in the experiment with simulated effective pressure from a hydrology model (RC_eN_GlaDS), we assumed a constant effective pressure, neglecting changes in basal drag over time. To address this, we conducted a separate experiment, RC_eN_GlaDS_HAF, by applying HAF-scaling to the simulated effective pressure.

Our results show that ice dynamics are highly sensitive to how temporally varying effective pressure is treated, particularly near the grounding line. This underlines the importance of using a coupled ice sheet–hydrology model, which would allow for a more realistic representation of the evolving hydraulic system, particularly under future projections when significant changes to basal conditions are expected. This has been well addressed in the Discussion (Line 212-219).

3. Structure of results and headlines

I believe the presentation of the results could be enhanced for better readability and comprehension. While the beginning that focuses on the implicit handling of N does a good job of describing the difference between LW and RC_iN_HAF, the following parts could be arranged in a more organized way, as I experienced them hard to follow. One idea could be to adapt something like the following structure (very similar to the current one):

Keep the first part on implicit handling of N: LW vs RC_iN_HAF as it is (L. 59 to L. 80). Then have a second part that covers the explicit handling of N: RC_eN_GlaDS vs RC_eN_POC vs RC_eN_GlaDS_HAF. Here I would suggest to start with RC_eN_POC as the most simple approach and then introduce GlaDS and the extended GlaDS (RC_eN_GlaDS_HAF).

With this continental view established, continue with the Tipping points and the regional analysis.

Not much change is needed, but these sections should be clearly identifiable and might deserve their own headings, also because the current in-between headings (“Subglacial effective pressure must be included in sliding relations” and “Subglacial effective pressure controls ice fluxes”) are very general and do not help much to navigate the text. The final part before the discussion (“Increased sensitivity to effective pressure near the grounding line”) is in a good place.

Many thanks for the suggestions, which are super helpful. We have reorganised the *Results* section as recommended:

Subglacial effective pressure must be included in sliding relations

This section compares the experiments using linear Weertman and Regularised Coulomb sliding relations and indicates that effective pressure is necessary in sliding relations.

Subglacial effective pressure controls ice fluxes

This section indicates the influence of different treatments of effective pressure on the projected mass loss. To maintain consistency with the structure of the *Introduction* section and adhere to Reviewer 2's suggestions, we presented the results using three bullet points:

- Implicit representation of effective pressure N
- Explicit representation of effective pressure N
- Basin-scale response

Increased sensitivity to effective pressure near the grounding line

This section emphasizes the critical role of effective pressure near the grounding line in affecting the low-drag area and projected ice mass loss.

We believe that the original subtitles effectively convey the key messages when supplemented by these key bullets. Please see the modified main text for detailed changes.

At some point, Kazmierczak et al. (2022) should also be referenced, as they also investigate the influence of different sliding relations for Antarctica.

Thanks for sharing this paper. We have cited this paper in our *Introduction* (Line 26-27) and *Discussion* (Line 213-215), especially about the finding that decreased effective pressure near the grounding line amplifies the ice sheet's sensitivity to climatic forcing, particularly for a given sliding relation power.

Remarks by lines

L. 29: also frictional heating due to internal deformation of ice

Thanks. Added.

L. 45f: Can't the effect of N be also approximated in the sliding coefficients for Weertman sliding laws? see also concern 1.

The short answer is “no”, except for the present-day state of the ice sheet. The effect of effective pressure N can be viewed as being implicitly represented in the spatial pattern of the Weertman coefficient, but this varies by many orders of magnitude because the Weertman sliding laws do not explicitly capture anything about the role of subglacial hydrology in sliding nor its evolution over time; all the relevant physics are hidden in one spatially tuned parameter. So we have no confidence in the contribution of N to projected evolution of the ice sheet when using Weertman sliding.

L. 48: more realistic than what?

Here we mean the HAF-scaling is more realistic than using a constant coefficient in RC_{iN} or a constant simulated effective pressure in RC_{eN_GlaDS} . This is because the

effective pressure is expected to vary with time, especially in regions where the ice approaches flotation with a moving grounding line. To address this and incorporate other related comments regarding the HAF-scaling, we have rewritten this entire paragraph as follows (Line 64-73):

“A further assumption about the effective pressure and its impact on sliding can be imposed using on "height above floatation" (HAF) scaling²¹. HAF is defined as the height of the upper ice surface above sea level minus the height required for the ice overburden pressure to match the ocean water pressure at the bed. This assumption scales the basal resistance or the effective pressure from a prescribed HAF value down to zero at the grounding line, reflecting the influence of warm seawater intrusion beneath grounded ice⁴¹. However, the inland extent of this effect is poorly constrained due to limited observations, and this approach neglects significant spatial and temporal variability driven by evolving glacier geometry^{42,43}. Both the HAF-scaling and perfect ocean connection assumption enforce the floating condition at the grounding line. However, HAF-scaling, applied only near the grounding line, adjusts basal resistance based on ice elevation relative to flotation, potentially under- or overestimating effective pressure. In contrast, the perfect ocean connection assumes uniform hydraulic connectivity across the domain, systematically overestimating effective pressure by neglecting hydraulic potential gradients.”

L. 56: Why SSA, what is the implication?

The SSA simplifies the full-Stokes equations by neglecting vertical shear stresses, focusing only on the horizontal stress balance. This reduction significantly decreases the computational cost, making SSA suitable for large-scale and long-term simulations, i.e. century-scale Antarctic Ice Sheet projections. The SSA is well-suited to simulate the behavior of fast-flowing ice streams, outlet glaciers, and floating ice shelves, which are critical regions influencing ice discharge from Antarctica. However, the neglect of vertical shear can lead to inaccuracies in simulating inland ice dynamics and interactions with the grounding zone, which might underestimate or oversimplify the response of grounded ice to changes in basal conditions or surface mass balance.

To address this limitation, we expanded the *Discussion* section to provide a more detailed explanation of the study's limitations (Line ***:

“Our simulations explore the sensitivity to basal sliding parameterisations under various climate scenarios. However, limitations in our approach highlight areas for improvement. The use of SSA assumes vertical hydrostatic equilibrium and neglects vertical shear stresses, which restricts its ability to fully capture the ice dynamics at the grounding line. This is particularly relevant in regions with pronounced bedrock

slopes, where non-hydrostatic effects and deviatoric (bridging) stresses—ignored by SSA become significant. While a fully resolved full-Stokes model, combined with a contact problem formulation for the lower ice surface, would be needed to consistently resolve these dynamics⁵¹, we argue that the sliding relation itself is less sensitive to the choice of stress representation in the ice-flow model. For instance, a similar sensitivity to driving stress has been found between an SSA model and one incorporating simple vertical shear, particularly under a regularized Coulomb law or a Weertman sliding parameterisation²¹.

Nevertheless, unresolved processes may further alter projected sea-level contributions. Given the importance of the grounding line region to our results, we expect further changes when considering three-dimensional shear fields, hydrology-ocean-bedrock-sediment interactions, iceberg calving, and uncertainties in predicted future surface mass balance. While these processes are unlikely to negate the effects we identify—given that effective pressure is largely isolated from external forcings—they underscore the complexity of grounding line dynamics and the need for urgent model development to advance the fidelity of sea-level projections.”

L. 57: Why is SSA used here? Why not hybrid/higher order? What are the implications?

See my response to L.56 above.

L. 60: Why is RC_iN_HAF more realistic? is combined effects of seawater intrusion and meltwater production is also considered in the eN variants? is this only about RC vs LW? more explanation is needed.

For the 1st and 3rd question here, Yes, the comparison here is only about RC vs LW. We suggest that RC relation is more physically realistic because it can capture both Weertman and Coulomb regimes, provides a unified framework for modelling sliding across diverse bed types as explained in the modified *Introduction* (Line 43-46). The HAF-scaling in RC_iN_HAF can well represent the combined effects of seawater intrusion and meltwater production near the grounding line as explained in Line 66-68. However, considering the restructured *Introduction* and *Results* sections, we deleted this sentence here and added a similar sentence in the *Introduction* (Line 77-80):

“For comparison against the more physically justifiable RC sliding parameterisation, we also apply the linear Weertman sliding relation (hereafter “LW”), which, despite its tendency to underestimate mass loss²¹, is still widely used for its simplicity in both numerical and computational aspects^{2, 5, 26, 30}.”

For the second question, while we agree that RC_eN_POC also considers the combined effects of seawater intrusion and meltwater production, it is important to note that RC_eN_GlaDS uses a constant effective pressure simulated from the GlaDS hydrology model. To account for a temporally evolving effective pressure, we applied HAF-scaling to the simulated effective pressure in RC_eN_GlaDS_HAF. We have further clarified the HAF scaling, along with its similarities and differences with POC, in the revised *Introduction* (Line 70-73).

L. 68: Replace “near mass balance” with “near equilibrium”. I don’t understand the second part of the sentence regarding observed mass loss.

Modified.

The second part of the sentence refers to how the RC relation suggests a near-zero mass loss by 2300, which aligns more closely with the observed mass balance trends in recent decades. We have revised the sentence for greater clarity, as follows (Line 91-93):

“Under the two low emission scenarios, the RC relation suggests near equilibrium by 2300, which aligns more closely with observed mass loss in recent decades⁴⁴, ...”

L.71: “For WAIS...”: Sentence is unclear. Maybe “For the WAIS the experiment using the RC relation and high emission scenario,...”

Modified.

L. 73: produce.

Modified.

L. 74: “However, the RC...” -> “However, using the RC...”

Modified.

L. 75: “than LW” -> “than using the LW relation”. The whole paragraph seems to be missing words.

Thanks for the suggestions. The whole paragraph has been modified accordingly (Line 95-105).

L. 82: This sentence is not very clear, seems obvious.

We modified the sentence to increase the clarity but we’re not sure what the reviewer is requesting here.

“Our simulations reveal the treatment of effective pressure significantly affects the spatial tuning of sliding coefficients, which, in turn, impacts the evolution of basal shear stress” (Line 111-112)

L 99ff: Isn't the addition of HAF to the GlaDS N a little bit of double counting, because the effect of the ocean is already included in the GlaDS model? Though it is only appropriate at the time where N was computed in GlaDS.

No, as we mentioned in response to an earlier comment, RC_eN_GlaDS uses a constant effective pressure simulated by the GlaDS hydrology model. While the GlaDS model does account for ocean effects, the constant simulated effective pressure becomes unsuitable for regions experiencing grounding line retreat. To address this time-varying effect, we apply HAF-scaling to the simulated effective pressure.

L. 106: Explain coulomb limit

Thanks for your comment. The Coulomb limit defines the maximum basal shear stress under a Coulomb sliding regime, where sliding is primarily controlled by effective pressure and the sliding coefficient. We decide to move all the sentences related with Coulomb limit into a separate paragraph in the subsection *“Increased sensitivity to effective pressure near the grounding line”* (Line 199-210). In the modified *Methods* section, we further explained the shift between Weertman and Coulomb regime under different assumptions of effective pressure (Line 475, 492).

We modified the sentence here into: *“While this parameterisation ensures the Coulomb regime dominates near the grounding line, ...”* (Line 131-132)

L. 115-122: Can the experiments using POC produce meaningful results for basins with reverse bed slope? Won't POC on the retrograde bed will lead to higher basal pressure in the inland than at the grounding line?

No, the POC assumption does not produce meaningful results, as it consistently underestimates water pressure inland. We included it in our study because it is one of the commonly used assumptions (see modified *Introduction*, Line 52-55, Line 72-73), not because we believe it to be a good assumption.

Yes, on a retrograde bed, POC will lead to higher basal water pressures inland compared to the grounding line, but these pressures are still insufficient. Given that basal friction and geothermal heat provide a basal water source nearly everywhere under the ice, and basal water sinks are rare, there is always a hydraulic gradient driving water toward the grounding line. In other words, subglacial water pressure is expected to build up sufficiently to drive flow toward the grounding line under normal conditions.

L. 153: How valid is the distribution of N computed by GlaDS for present day conditions after more than a century? Even with the HAF scaling, the whole hydraulic regime might have shifted.

The distribution of N computed by GlaDS for present-day conditions is based on the initial geometry and the friction heating calculated from the inverted basal shear stress. We acknowledge that over more than a century, changes in the ice sheet and surrounding environment—such as variations in oceanic conditions, climate, and ice dynamics—could have altered the hydraulic regime, especially near the moving grounding line. While the HAF-scaling helps account for some of these time-varying effects, it is true that the entire hydraulic regime may have shifted, particularly in regions where significant changes in the ice flow or grounding line dynamics have occurred. However, in this study, we're not aiming to provide an accurate representation of the evolving hydrology over time, which can only be achieved by applying a coupled ice sheet-hydrology model. We have made it clear at the start of the Methods section (Line 414-416):

“Our simulations do not capture a fully coupled ice-hydrology system, as such a model is not yet available for continental-scale applications. Instead, our simulations aim to determine whether a coupling is indeed necessary, rather than predetermining what might be considered a valid future projection of ice mass change.”

Moreover, we emphasized the limitation of HAF-scaling and the importance of having a coupled ice-hydrology model in the modified Discussion (Line 221-229):

“Among these RC experiments, we place greater confidence in the results from experiment RC_eN_GlaDS_HAF, as it uses a more physically realistic subglacial hydrology system where HAF-scaling adjusts effective pressure as the grounding line retreats. However, while the HAF-scaling accounts for some of these time-varying effects, the entire hydraulic regime may have shifted, particularly in regions experiencing significant changes in ice flow or grounding line dynamics^{12, 39}. Moreover, the GlaDS model, originally designed to simulate subglacial hydrology on hard bedrock, may predict more efficient drainage systems than those occurring on a soft bed, potentially leading to lower modelled subglacial water pressures^{12,37,39}. This highlights the importance of accurately predicting subglacial water pressure and developing a more comprehensive ice sheet-hydrology model that can account for the temporal co-evolution of ice sheet dynamics and the subglacial hydrological system.”

L. 163: What does “constrained future surface mass budgets estimates” mean? Improvement of the predictions of SMB?

Yes, by "constrained future surface mass budget estimates," we mean the uncertainties in predicting the future surface mass balance (SMB). To clarify, we have revised the sentence to: “...and uncertainties in predicted future surface mass balance.” (Line 241)

Figure 1: good colors, bigger fonts, no box around legend

Thanks for the suggestions. Based on comments from Reviewer 1, we have modified Figure 1 to be more colour-blind friendly with bigger fonts and no box around legend.

Figure 2:

caption: “sea-level contributions between linear Weertman law” -> “sea-level contributions in experiment using linear Weertman law”.

Modified.

Figure 4:

caption: “with a Hydrology model” -> “from GlaDS”

Modified.

Does channel area mean cross section of individual channels?

Yes, the channel area refers to the cross section of each channel. To clarify this, we have updated the caption of Fig. 4 to read: “The background image shows the simulated effective pressure and channel area (cross-sectional area of each channel) generated from GlaDS.”

why is a diverging colormap used? What is special about 0.5 MPa?

The diverging colormap and the value of 0.5 MPa do not have any special significance. We chose this color scale to emphasize regions of low effective pressure, such as the Siple Coast and Lake Vostok.

small inset figures in G-H and D-Dp are hardly readable.

We have zoomed in the inset figures for G-H and D-Dp.

Figure 5:

Why are the lines dashed?

There is no special meaning here. We have modified them into solid lines.

References

DE FLEURIAN B, WERDER MA, BEYER S, et al. SHMIP The subglacial hydrology model intercomparison Project. *Journal of Glaciology*. 2018;64(248):897-916. doi:10.1017/jog.2018.78

Kazmierczak, E., Sun, S., Coulon, V., and Pattyn, F.: Subglacial hydrology modulates basal sliding response of the Antarctic ice sheet to climate forcing, *The Cryosphere*, 16, 4537–4552, <https://doi.org/10.5194/tc-16-4537-2022>, 2022.

Bueler, E., and J. Brown (2009), Shallow shelf approximation as a “sliding law” in a thermomechanically coupled ice sheet model, *J. Geophys. Res.*, 114, F03008, doi:10.1029/2008JF001179.

Morlighem, M., E. Rignot, H. Seroussi, E. Larour, H. Ben Dhia, and D. Aubry (2010), Spatial patterns of basal drag inferred using control methods from a full-Stokes and simpler models for Pine Island Glacier, West Antarctica, *Geophys. Res. Lett.*, 37, L14502, doi:10.1029/2010GL043853.

Reviewer #3 (Remarks to the Author):

Review of “Subglacial Water Amplifies Antarctic Contributions to Sea-Level Rise”, for Nature Communications, by Zhao, Gladstone, Zwinger, Gillet-Chaulet, Wang, Caillet, Mathiot, Saraste, Jager, Galton-Fenzi, Christoffersen and King

This manuscript describes the importance of incorporating subglacial hydrology parameterizations into whole Antarctic ice sheet models, by varying parameterization of basal water effective pressure in a basal slip relation. The authors conclude that effective pressure is required in the slip relationship, and that ice flux is sensitive to effective pressure model, especially near the grounding line.

We believe the topic of this manuscript is important and can be of interest to a wide range of readers. However, we believe that this manuscript can be significantly improved by considering the comments listed below.

For now, the manuscript is “expert-facing”, i.e., it is assumed that the readers have a substantial understanding of the background and methods of this study. The manuscript leaps over many (essential) details that could have been helpful for the readers to follow along.

We have gone through the manuscript line by line with a view to providing clarity for non-experts. This is most noticeable in the modified *Introduction*. See also our specific responses to further comments below.

- For an important study that can be of interest to the general public, especially in the journal of Nature Communication, we argue this is not appropriate. We recommend the authors to provide more detailed elaborations on the background, methods, findings, and implications of this study. A few examples can be found in the comments below. Thanks for the comments. We have further elaborated on these sections. See our specific responses to further comments below.

- In the section “Subglacial effective pressure must be included in sliding relations”; its implied the observed mass gain of the linear Weertman models demonstrate the need for subglacial effective pressure. The authors need to make a stronger case - ie demonstrate that mass gain is not plausible over these time frames.

The comparison between linear Weertman (LW) and regularised Coulomb (RC) sliding relations suggests that inclusion of effective pressure in the sliding relation largely enhances the basal sliding and projected mass loss. Whether the mass gain simulated

in LW is plausible is not the focus here. We have clarified this in the modified *Methods* section (Line 414-416) that we do not aim to predetermine what might be considered a valid future projection in terms of mass change. We have also made minor modifications to the *Introduction* section (Line 77-80) to make it clear that the existing literature already provides strong evidence for favouring RC over LW. The existence of a wide spread hydrological network under the Antarctic Ice Sheet can lubricate the ice base and thus lead to increased ice velocities, which should be considered in the basal sliding relations. Our aim in the current study is not to convince the reader that RC is better than LW (this has already been done), but rather to show just how large an impact this choice can have.

- We recommend the authors to elaborate on the differences between model setups. We appreciate the balance between the amount of work presented in this manuscript and the space limitations. But the current manuscript makes it really difficult to follow the different model setups, especially considering the number of runs that are being presented and discussed.

Thanks for the suggestions. We have provided a more accessible conceptual description of the different approaches in the modified *Introduction* section, i.e. the different assumptions implied by the different versions of sliding laws. We have also moved the modified **Table 1** to the *Introduction* to help better interpret the differences. See our response to Reviewer 2 for more details.

- It is not clear why/how including different subglacial hydrology components can change the AIS SLR projection by such a large degree. What is the physical process behind the differences? We recommend the authors to elaborate on this and make it more clear.

Thanks for the comment. We have added a description of the most important physical processes at the start of the subsection “*Increased sensitivity to effective pressure near the grounding line*” (Line 174-182). The key concept, which we now believe comes across clearly to the non-expert, is that of a low friction “ice plain” immediately upstream of the grounding line. This section has also been expanded to include a discussion of how assumptions about effective pressure influence the extent of this low-friction “ice plain” region (Line 262-273), which can be widely different depending on the bedrock and ice geometry.

- Line 47-52: HAF is a very important concept for this study. However, the authors didn’t provide any explanation/elaboration on the concept, other than providing one reference. (This is also true for some other important concepts throughout the manuscript.) This approach makes it difficult for non-expert readers to understand the work and follow the logic flow. We recommend providing a more detailed explanation of the concept, and/or maybe even a conceptual diagram.

Thanks for the comment. We have provided an accessible description of the concept of HAF in the modified *Introduction* (Line 65-68):

“HAF is defined as the height of the upper ice surface above sea level minus the height required for the ice overburden pressure to match the ocean water pressure at the bed. This assumption scales the basal resistance or the effective pressure from a prescribed HAF value down to zero at the grounding line, reflecting the influence of warm seawater intrusion beneath grounded ice ⁴¹.”

We have also provided a more detailed explanation for other concepts like the Coulomb limit (Line 201-202)

- It is not clear what assumption is made about the subglacial meltwater budget. Does the model run only require a certain and finite amount of subglacial water? Or do the authors assume an “unlimited” supply of subglacial water generated by basal melting upstream? And does a limited subglacial meltwater budget change the model output?

Thanks for the comments. We have expanded our description of the hydrology model in the *Methods* section (Line 511-519). The hydrology model spin-up does make specific assumptions about the basal water source and the other parameters on which basal water pressure depends, most important of which are parameters describing conductivity of both basal channels and the distributed basal water system. A brief description is given in the *Methods* section (Line 515-516), and the citation to the recent paper (Zhang et al., 2024) is the most directly relevant as their use of GlaDS is almost identical to the current study.

The ice dynamic simulations have no direct dependence on water budget; instead, they depend on basal water pressure. This pressure is influenced by various factors, particularly the conductivities mentioned above, with basal water production being driven solely by frictional heating. Consequently, the only simulations where assumptions about the basal water budget are directly relevant are those utilizing the spatial distribution of basal water from the hydrology model. Even in those cases, the dependence remains weak due to the influence of other factors.

To answer the question literally (though the fact that our hydrology model is in steady state probably makes this clarification unnecessary), the subglacial water supply is effectively unlimited, but any other answer would result in no hydrology at all at steady state.

The feedback between ice dynamics and basal water generation, through friction heating due to sliding, may be important for future evolution of the coupled system, but this is beyond the scope of the current study.

* The authors need to address the limitation of the study in the Discussion section. We understand that the authors need to consider the space limitations per journal guideline, but we strongly recommend the authors to reconsider the balance between (1) further elaboration and discussion of the study findings, (2) connection with other studies, (3) broader implications, and (4) limitation of the study, for the Discussion section.

Thanks for your comments. We have carefully considered your suggestions and revised the *Discussion* section, which we hope will satisfy this request. Specifically:

1. **Elaboration and discussion of findings:** We have added a paragraph emphasizing the critical role of subglacial water pressure and the significant influence of a small region near the grounding line on multi-century ice sheet mass loss. This discussion highlights the connection between effective pressure and sliding relations, as well as the significance of model resolution in capturing small-scale transitions near the grounding line (Lines 212–219).
2. **Connection with other studies:** We have incorporated relevant citations in the *Discussion* to place our findings in the context of previous research. For example, we cite Kazmierczak et al. (2022) to emphasize the influence of different sliding relations for Antarctica (Lines 213–215), suggested by Reviewer 2.
3. **Broader implications:** In the final paragraph of the *Discussion* (Lines 245–251), we elaborate on the implications of our findings for understanding ice sheet dynamics and their influence on sea-level projections. This paragraph emphasizes the importance of integrating realistic subglacial processes into future models to improve predictions.
4. **Study limitations:** In response to your comments and those of Reviewer 2, we have added a paragraph discussing the limitations of using the Shallow Shelf Approximation (SSA) in this study (Line 230-238). Additionally, the original submission already addressed other limitations, such as hydrology-ocean-bedrock-sediment interactions, iceberg calving, and uncertainties in future surface mass balance projections (Line 239-241).

minor comments:

* This manuscript uses a lot of acronyms - are they really necessary? Like, “GL” for “grounding line”? Having too many acronyms can be counterproductive. Also, it may help to spell out the run names in Figure 1.

Thanks for the comments. We have removed the acronyms including GL (grounding line), N (effective pressure), SLR (sea-level rise), AIS (Antarctic Ice Sheet), WAIS (West

Antarctic Ice Sheet), EAIS (East Antarctic Ice Sheet). We now only keep three acronyms: HAF (height above flotation), LW (linear Weertman), RC (regularised Coulomb).

We have moved the modified Table 1 from *Methods* to *Introduction*, which can well explain the difference across these experiments in Figure 1.

* We noticed that no doi/web-link is provided for the references. We encourage the authors to consider including such information.

Our manuscript is available in LaTeX format, which includes the DOI and web links in the references. If the manuscript is accepted, we hope the editing team can assist with ensuring the DOI and web links are properly included in the final references.

Unsure of what is meant by the second and third authors contributed equally.

We mean that the second and third authors contributed to this paper equally. To reduce the confusion, we removed this statement.

Reviewer #4 (Remarks to the Author):

Many thanks for your contribution to provide the valuable comments.

References:

- Dow, C.F., Ross, N., Jeofry, H. et al. Antarctic basal environment shaped by high-pressure flow through a subglacial river system. *Nat. Geosci.* 15, 892–898 (2022). <https://doi.org/10.1038/s41561-022-01059-1>
- GAGLIARDINI, O. and WERDER, M.A. (2018) 'Influence of increasing surface melt over decadal timescales on land-terminating Greenland-type outlet glaciers', *Journal of Glaciology*, 64(247), pp. 700–710. doi:10.1017/jog.2018.59.
- Joughin, I., Smith, B. E., & Schoof, C. G. (2019). Regularized Coulomb friction laws for ice sheet sliding: Application to Pine Island Glacier, Antarctica. *Geophysical Research Letters*, 46, 4764–4771. <https://doi.org/10.1029/2019GL082526>
- Pelle, T., Greenbaum, J. S., Ehrenfeucht, S., Dow, C. F., & McCormack, F. S. (2024). Subglacial discharge accelerates dynamic retreat of aurora subglacial basin outlet glaciers, East Antarctica, over the 21st century. *Journal of Geophysical Research: Earth Surface*, 129, e2023JF007513. <https://doi.org/10.1029/2023JF007513>
- Seroussi, H. and Morlighem, M.: Representation of basal melting at the grounding line in ice flow models, *The Cryosphere*, 12, 3085–3096, <https://doi.org/10.5194/tc-12-3085-2018>, 2018.
- Seroussi, H., Pelle, T., Lipscomb, W. H., Abe-Ouchi, A., Albrecht, T., Alvarez-Solas, J., et al. (2024). Evolution of the Antarctic Ice Sheet over the next three centuries from an ISMIP6 model ensemble. *Earth's Future*, 12, e2024EF004561. <https://doi.org/10.1029/2024EF004561>
- Wang, Y., Zhao, C., Gladstone, R., Zwinger, T., Galton-Fenzi, B. K., and Christoffersen, P.: Sensitivity of the future evolution of the Wilkes Subglacial Basin ice sheet to grounding-line melt parameterizations, *The Cryosphere*, 18, 5117–5137, <https://doi.org/10.5194/tc-18-5117-2024>, 2024.
- Weertman, J. (1972). *General theory of ice-sheet flow*. *Journal of Glaciology*, 11(62), 157-170. DOI: 10.1017/S002214300002302X
- Werder, M. A., I. J. Hewitt, C. G. Schoof, and G. E. Flowers (2013), Modeling channelized and distributed subglacial drainage in two dimensions, *J. Geophys. Res. Earth Surf.*, 118, doi:10.1002/jgrf.20146.
- Yufang Zhang, John C Moore, Liyun Zhao, Mauro A Werder, Rupert Gladstone, and Michael Wolovick. The role of hydraulic conductivity in the Pine Island Glacier's subglacial water distribution. *Science of The Total Environment*, 927:172144, 2024.

Response to Reviewers

Many thanks to the reviewers for their valuable comments. Our responses are highlighted in blue, and the line numbers refer to the revised manuscript.

Reviewer #1 (Remarks to the Author):

The paper is the result of a numerical study investigating the effect of basal pressure under the Antarctic ice sheet and the role of the pressurized water layer in basal slip. While for a direct prognostic application there is lack of data, the paper defines the range of solutions for Ice Sheet/iceshelf dynamics depending on pressure in particular at the icesheet-iceshelf boundary.

The authors have responded to all reviewers comments and I am satisfied with the answers to my specific comments.

Thanks again for your comments.

Reviewer #2 (Remarks to the Author):

The authors have addressed the reviewers' concerns effectively. I'm pleased with the improved paper structure and extended description. I have only one minor comment:

I appreciate the more detailed description of sliding laws used in the ISMIP/ABUMIP experiments, but I would like to point out that the PISM model uses a Mohr-Coulomb type sliding law and not a Weertman type. It is specifically mentioned in the manual that Weertman should not be used with PISM: <https://www.pism.io/docs/manual/modeling-choices/dynamics/weertman.html> and also in the model description it is pretty clear:

Seroussi et al. 2020

PIK_PISM

We apply a power law for sliding with a Mohr–Coulomb criterion relating the yield stress to parameterized till material properties and the effective pressure of the overlaying ice on the saturated till (Bueler and van Pelt, 2015).

Seroussi et al. 2024:

PIK_PISM

A generalized power law (Schoof & Hindmarsh, 2010) is applied to parameterize basal sliding. The basal friction coefficient depends on the effective pressure and till friction angle, that is parameterized using a heuristic, piecewise linear function of the bed elevation (Martin et al., 2011).

I know from personal communication that AWI_PISM uses the same setup in that regard.

This does of course not change the overall conclusion of the authors that Weertman type sliding laws are dominant in the benchmarks.

Many thanks for the correction and providing the detailed information. I have modified the Table S1 in the supplementary material accordingly. See the modified table below.

ISMIP6-2100 ⁶		ISMIP6-2300 ⁷		ABUMIP ⁸	
Model Name	Sliding Law	Model Name	Sliding Law	Model Name	Sliding Law
AWI_PISM	Mohr-Coulomb (m=3)	-	-	AWI_PISMPal	Coulomb
-	-	UCM_Yelmo	Regularised Coulomb	-	-
IMAU_IMAUICE1	Weertman(m=3)	MAU_UFEMISM	Regularised Coulomb	IMAU_ICE	Coulomb
IMAU_IMAUICE2	Weertman(m=3)	-	-	-	-
NCAR_CISM	Weertman(m=3)-Coulomb	NCAR_CISM1	Weertman(m=3)-Coulomb	NCAR_CISM	Weertman(m=3)-Coulomb
-	-	NCAR_CISM2	Zoet-Iversen	-	-
-	-	NORCE_CISM	Zoet-Iversen	-	-
DOE_MALI	Weertman(m=1)	DOE_MALI	Weertman (m=3)	DOE_MALI	Weertman(m=1)
ILTS_PIK_SICOPOLIS	Weertman(m=3)-Budd	ILTS_PIK_SICOPOLIS	Weertman(m=3)-Budd	ILTS_PIK_SICOPOLIS	Weertman(m=3)
JPL_ISSM	Budd(m=1)	UCSD_ISSM	Weertman(m=3)-Budd	JPL_ISSM	Weertman(m=1)
UCIPL_ISSM	Weertman(m=3)	-	-	-	-
PIK_PISM1	Mohr-Coulomb(m=3)	PIK_PISM	Mohr-Coulomb (m=3)	PIK_PISM	Coulomb
PIK_PISM2	Mohr-Coulomb (m=3)	-	-	-	-
ULB_FETISH_16km	Weertman(m=2)	ULB_FETISH	Weertman(m=3)	ULB_FETISH	Weertman(m=2)
ULB_FETISH_32km	Weertman(m=2)	-	-	-	-
UTAS_ELMERICE	Weertman(m=1)	UTAS_ELMERICE	Weertman (m=1)	-	-
-	-	IGE_ElmerIce	Weertman (m=1)	IGE_ElmerIce	Weertman (m=3)
VUB_AISMPALEO	Weertman(m=3)	-	-	-	-
VUW_PISM	Weertman(m=3)	VUW_PISM1	Weertman(m=3)	-	-
-	-	VUW_PISM2	Weertman(m=3)	-	-
-	-	-	-	ARC_PISM1	Coulomb
-	-	-	-	ARC_PISM2	Coulomb
LCSE_GRISLI	Weertman(m=3)	LCSE_GRISLI	Weertman(m=3)	LCSE_GRISLI	Coulomb
-	-	LCSE_GRISLI2	Weertman(m=3)	-	-
-	-	-	-	CPOM_BISICLES	Weertman(m=3)-Coulomb
-	-	-	-	PSU_PSU3D1	Weertman(m=2)
-	-	-	-	PSU_PSU3D2	Weertman(m=2)

Table S1. Sliding law used in ISMIP6-2100 Antarctic⁶, ISMIP6-2300 Antarctic⁷, and ABUMIP⁸.

As the reviewer noted, this does not change our conclusion stated in the *Introduction* “the majority of Antarctic Ice Sheet models employed the Weertman sliding relation”. To ensure statistical accuracy, we have revised this sentence to:

“A summary of sliding laws used in three major Model Intercomparison Projects (Table S1) reveals that the majority of Antarctic Ice Sheet models (e.g., 10 of 16 in ISMIP6-2100², 8 of 16 in ISMIP6-2300⁵, and 7 of 15 in ABUMIP³⁰) employed the Weertman sliding relation.” (Line 49-51)

Reviewer #3 (Remarks to the Author):

This work illustrates the need for improved knowledge of the evolution of subglacial hydrology, and we recommend publication.

The revised manuscript reads much better with acronym expansion and with more detailed concept explanations and discussions. Our only minor follow-up suggestion is to explicitly state in the Method section that the subglacial water budget is held constant because (1) the model requires such assumption to reach steady state, and (2) different assumptions on this do not cause significant variation in the modeling results. Many thanks for the suggestions. We have added this statement in the *Methods* section: **“To ensure that the hydrology system reaches a steady state, the total subglacial water budget is held constant throughout the spin-up period (over approximately 100 model years). This assumption introduces negligible variability in the final results compared to alternative assumptions on subglacial melt supply.”** (Line 519-521)

Reviewer #4 (Remarks to the Author):

Many thanks for your contribution to provide the valuable comments.